# Specific quantification of inducible HIV-1 reservoir by RT-LAMP
Tanvir Hossain [1,8], Cynthia Lungu [1,8], Sten de Schrijver [1], Mamokoena Kuali[2], Raquel Crespo[1], Nicole Reddy[3], Ayanda Ngubane[2], Tsung Wai Kan[4,5], Kavidha Reddy[3], Shringar Rao[1], Robert-Jan Palstra[4,5], Paradise Madlala[2], Thumbi Ndung'u[2,3,6,7] & Tokameh Mahmoudi [1,4,5] ✉

## Abstract

**Background** Strategies toward HIV-1 cure aim to clear, inactivate, reduce, or immunologically control the virus from a pool of latently infected cells such that combination antiretroviral therapy (cART) can be safely interrupted. In order to assess the impact of any putative curative interventions on the size and inducibility of the latent HIV-1 reservoir, robust and scalable assays are needed to precisely quantify the frequency of infected cells containing inducible HIV-1.

**Methods** We developed Specific Quantification of Inducible HIV−1 by RT-LAMP (SQuHIVLa), leveraging the high sensitivity and specificity of RT-LAMP, performed in a single reaction, to detect and quantify cells expressing *tat/rev* HIV-1 multiply spliced RNA (msRNA) upon activation. The LAMP primer/probe used in SQuHIVLa was designed to exclusively detect HIV-1 *tat/rev* msRNA and adapted for different HIV-1 subtypes.

**Results** Using SQuHIVLa, we successfully quantify the inducible viral reservoir in CD4+ T cells from people living with HIV-1 subtypes B and C on cART. The assay demonstrates high sensitivity, specificity, and reproducibility.

**Conclusions** SQuHIVLa offers a high throughput, scalable, and specific HIV-1 reservoir quantification tool that is amenable to resource-limited settings. This assay poses remarkable potential in facilitating the evaluation of potential interventional strategies toward achieving HIV-1 cure.

## Plain language summary

HIV infection remains challenging because the virus hides in certain cells, making it invisible to the immune system. This hidden virus forms what is called a latent HIV reservoir. If someone with HIV stops their antiviral therapy, the virus quickly re-emerges. Because of this, researchers are exploring various strategies to eliminate this reservoir and cure HIV. To evaluate these strategies, we need a method to measure the reservoir's size before and after trials. Our study introduces SQuHIVLa, a highly sensitive and specific method for quantifying the latent reservoir. SQuHIVLa could become a vital tool for monitoring HIV patients and assessing treatment effectiveness, bringing us closer to finding a cure.

Combination antiretroviral therapy (cART) has changed human immunodeficiency virus type 1 (HIV-1) infection from a lethal to a chronic illness. However, despite over 40 years of research, there is currently no safe, scalable, curative therapy in sight. Furthermore, HIV-1 treatment disparities persist, notably in sub-Saharan Africa, where 70% of individuals with HIV-1 reside[1], and only 78% of them with access to treatment[2]. Although cART effectively reduces circulating virus to undetectable levels, the integrated provirus persists within immune cells, primarily resting memory CD4+ T cells, constituting the viral reservoir[3,4]. Discontinuation of suppressive cART may prompt reactivation of the inducible replication-competent HIV-1 provirus, resulting in exponential replication and detectable viremia (viral rebound). This typically occurs within several weeks, with a mean of two

weeks, in a majority of individuals[5]. Lifelong therapy is thus necessary to maintain viral suppression and interventions beyond cART are urgently needed to achieve an HIV-1 cure.

Toward this aim, several strategies that target the latent viral reservoir for clearance or control are being explored[6,7]. Reliable biomarkers are crucial to understanding reservoir persistence and evaluating the impact of interventions. Furthermore, robust, precise, and high throughput and, scalable assays capable of measuring the frequency of infected cells containing inducible HIV-1 are required[8]. However, the low frequency of these cells, estimated at between 1 and 1000 per million CD4+ T cells[9] makes quantification extremely challenging, highlighting the need for innovative solutions to identify, target and quantify the viral reservoir.

[1]Department of Biochemistry, Erasmus University Medical Center, Rotterdam, The Netherlands. [2]HIV Pathogenesis Programme, The Doris Duke Medical Research Institute, University of KwaZulu-Natal, Durban, South Africa. [3]Africa Health Research Institute, Durban, South Africa. [4]Department of Urology, Erasmus University Medical Center, Rotterdam, The Netherlands. [5]Department of Pathology, Erasmus University Medical Center, Rotterdam, The Netherlands. [6]Ragon Institute of Massachusetts General Hospital, Massachusetts Institute of Technology and Harvard University, Boston, MA, USA. [7]Division of Infection and Immunity, University College London, London, UK. [8]These authors contributed equally: Tanvir Hossain, Cynthia Lungu. ✉e-mail: t.mahmoudi@erasmusmc.nl

Numerous techniques have been proposed, developed, or used to measure the HIV-1 reservoir[10 11,12]. Quantitative viral outgrowth assays that measure the replication-competent viral reservoir however underestimate the size[13,14]. PCR-based assays offer a more practical method to determine the frequency of cells containing total or integrated HIV-1 DNA[15,16]. However, these assays considerably overestimate the size of the viral reservoir. By targeting a single highly conserved proviral region, defective HIV-1 genomes with substantial deletions or hypermutations in places other than the test amplicon can also be detected[13,17]. Recent advancements address this challenge by introducing assays detecting multiple conserved HIV-1 genome regions. The intact proviral DNA assay (IPDA)[18], for example, targets the packaging signal (PSI) and env-RRE regions using digital droplet (dd)PCR technology. Although conventional IPDA is effective in excluding the majority of defective viruses, it targets a very small sub-genomic region (2% of the HIV-1 genome), which can lead to an overestimation of the viral reservoir[19]. Adapted multiplex versions of IPDA include additional targets[20,21] whereas some methods like Q4PCR, combining quadruplex qPCR and next-generation sequencing[22]. While these assays have revealed novel insights into HIV-1 proviral landscape dynamics, they are expensive, labor-intensive, and require sophisticated equipment, making them unsuitable for use in large-scale clinical trials, especially in resource-limited settings. Additionally, they exhibit high fail rates due to HIV-1 sequence polymorphisms[23,24] and near-full-length genome sequencing methods may introduce a quantification bias[25]. Importantly, proviral DNA sequence intactness alone is not a direct measure of inducibility, which greatly depends on the site of viral integration and therein the potential for viral transcription[13,26-32].

Novel culture-based techniques have emerged as potential alternates for estimating the inducible viral reservoir size. These methods involve the activation of CD4+ T cells from virally suppressed individuals, allowing direct measurement of HIV-1 RNA from cell extracts (cell-associated RNA, ca-RNA)[33-36] or cell culture supernatants (cell-free RNA, cf-RNA)[33,34]. However, cf-RNA quantification may not accurately assess replication-competency as it only reflects the capacity to generate and release viral RNA[33,37]. Inducible ca-RNA assays involve the potent activation of CD4+ T cells, enabling ultrasensitive PCR quantification of different RNA species including unspliced RNA (usRNA)[33,38], multiply spliced RNA (msRNA)[35,36,39], mature RNA transcripts with poly-A tails[38,40], TAR RNAs[41], and chimeric host-HIV-1 transcripts[38,39] can be quantified by RT-qPCR or RT-ddPCR methods. Since tat/rev msRNA transcripts are generated after splicing of full-length viral transcripts, they have been found to be a meaningful indicator of viral replication following latency reversal[39,42-45]. By detecting tat/rev msRNA therefore, the likelihood of measuring proviruses with large internal deletions is greatly reduced[13]. Inducible HIV-1 RNA assays can be conducted in a limiting dilution format, such as the tat/rev Limiting Dilution Assay (TILDA)[35,46], or at single-cell level by, for example, detection of HIV-1 usRNA and/or msRNA using fluorescent in situ hybridization assays coupled to flow cytometry (FISH-flow)[47,48]. While FISH-flow provides a deeper understanding of the viral reservoir molecular characteristics and phenotypic heterogeneity[47-49] it still requires substantial cell input and time, making it less suitable for large-scale clinical trials, favoring the use of bulk or serial-diluted sample measurements using quantitative real-time or digital PCR methods in most clinical studies[50].

HIV-1 RNA quantification typically relies on single-round or semi-nested RT-qPCR, with the latter offering advantages for low-copy samples and a wider dynamic range[39]. However, quantifying HIV-1 RNA in limiting dilution formats considerably increases the cost. Moreover, the instrument-intensive amplification procedure, long turnaround time, and high risk of cross-contamination from manual handling of the pre-amplified product in semi-nested RT-qPCR protocols, limit assay application in large clinical trials, especially in resource-constrained settings[51].

To address these technical limitations, reverse transcription loop-mediated isothermal amplification (RT-LAMP) could be used as a highly sensitive, less expensive, and time-saving alternative to semi-nested RT-qPCR[52]. Utilizing two polymerase enzymes, RT-LAMP achieves reverse transcription and amplification in a single reaction, minimizing cross-contamination risks. RT-LAMP has been successfully used in multiple studies to detect low copies of viral RNA for SARS-CoV-2 diagnosis, demonstrating comparable or better sensitivity than conventional RT-qPCR[53-56]. Additionally, by binding six to eight different DNA regions, RT-LAMP is more specific than semi-nested PCR, which targets three regions. Although RT-LAMP has the potential to be a less expensive and robust alternative to (semi-nested) RT-qPCR or RT-ddPCR[57], it has thus far only been used for qualitative HIV-1 RNA detection in blood plasma[58-60] and not yet in the context of HIV-1 reservoir quantification. Here, we present SQuHIVLa (Specific Quantification of Inducible HIV-1 reservoir by RT-LAMP), an RT-LAMP-based assay to detect cells expressing tat/rev msRNA, addressing the gap in HIV-1 reservoir quantification. We successfully designed LAMP primer/probe sets specifically for detecting tat/rev msRNA of HIV-1 subtypes B and C, while ensuring no amplification occurs from intron-containing tat/rev HIV-1 proviral DNA. Extensive experimental validations confirm that this innovative RT-LAMP based assay, named SQuHIVLa, demonstrates high sensitivity and specificity in exclusively amplifying tat/rev msRNA, minimizing the probability of false positives. Finally, we successfully quantify the frequency of tat/rev msRNA-expressing cells as a surrogate marker for inducible HIV-1 reservoir size using SQuHIVLa in CD4+ T cells obtained from PLWH with various clinical characteristics. Therefore, SQuHIVLa provides a robust and scalable tool for accurately assessing the size and inducibility of latent HIV-1 reservoirs, which is crucial for evaluating the efficacy of potential HIV-1 cure interventions.

## Methods
### Cohort characteristics
Peripheral blood samples from 34 people living with HIV-1 (PLWH), 15 subtype B, and 19 subtype C were used in this study. All 34 PLWH included in this study were older than 18 years of age and on fully suppressive combination antiretroviral therapy (cART), initiated during chronic stages of HIV-1 infection, with plasma HIV-1 RNA below detectable levels for at least twelve months. Additional clinical information is shown in Table 1 and Supplementary Table 5.

### Ethical statement
Written informed consent was obtained from all study participants. The Medical Ethical Council of the Erasmus Medical Center (MEC-2012–583) and the University of KwaZulu-Natal (REF: E036/06) approved the use of clinical material for research purposes.

### Sample collection
Peripheral blood mononuclear cells (PBMCs) were isolated from whole blood or leukapheresis samples using Ficoll density gradient centrifugation and cryopreserved in liquid nitrogen until further use. CD4+ T cells were isolated from thawed PBMCs by negative magnetic selection using the EasySep Human CD4 T Cell Enrichment Kit (STEMCELL Technologies) according to the manufacturer's protocol.

### Designing of tat/rev RNA-specific LAMP primer/probe sets
A detailed protocol for designing HIV-1 tat/rev msRNA-specific LAMP primers and probes is provided in Supplementary Note 1. Briefly, ten complete HIV-1 genome sequences for both HIV-1 subtype B and subtype C, submitted from various geographical locations and from different years, were obtained from the Los Alamos HIV sequence databases (https://www.hiv.lanl.gov/)[61] (GenBank Accession numbers of the sequences used to design subtype B and C tat/rev msRNA LAMP primers are listed in Supplementary Table 1). The sequences were aligned using the Molecular Evolutionary Genetics Analysis software version 11 (MEGA11)[62]. A consensus in silico-spliced tat/rev DNA sequence was generated using this alignment, converted to a FASTA file, and used to generate preliminary sets of LAMP primers using the online program PrimerExplorer V5 (http://primerexplorer.jp/e/). Selected LAMP primers, based on criteria that we

**Table 1 | Participant characteristics of HPP participants (HIV-1 subtype C)**

| Patient ID | Age (years) | Sex assigned at birth | ART regimen | Viral load at sample collection | CD4+ T cell count (cells/mm³) | Duration of viral suppression (months) | Duration of infection before ART initiation (months) | CD4 nadir (cells/mm³) | Pre-cART plasma HIV-1 RNA (log10 copies/mL) | Reservoir size (SQuHIVLa) (cells/millions of CD4+ T cells) | Total HIV DNA (ddPCR) (copies/millions of CD4+ T cells) |
|---|---|---|---|---|---|---|---|---|---|---|---|
| HPP 37 | 21–25 | F | ODIMUNE | <20 | 751 | 26–30 | 18 | 429 | 3,00 | 57,53 | NA |
| HPP 39 | 25–30 | F | ATROIZA | <20 | 953 | 21–25 | 12 | 497 | 4,76 | 38,9 | NA |
| HPP 46 | 21–25 | F | TRIBUSS | <20 | 708 | 21–25 | 15 | 307 | 3,85 | 87,54 | 271.46 |
| HPP 50 | 26–30 | F | TRIBUS | <20 | 855 | 26–30 | 16 | 524 | 4,36 | 48,47 | 472.81 |
| HPP 66 | 21–25 | M | ATROIZA | <20 | 1103 | 11–15 | 3 | 624 | 3,72 | 10,12 | 539.08 |
| HPP 70 | 18–20 | F | ATROIZA | <20 | 729 | 16–20 | 15 | 385 | 4,04 | 68,87 | 463.77 |
| HPP 71 | 21–25 | M | TRIBUS | <20 | 734 | 16–20 | <1 | 473 | 4,28 | 14,04 | 349.04 |
| HPP 73 | 31–35 | M | ATROIZA | <20 | 711 | 41–45 | 23 | 565 | 3,36 | 42,04 | 241.11 |
| HPP 75 | 21–25 | F | ATROIZA | <20 | 927 | 21–25 | 18 | 573 | 3,65 | 47,08 | 442.97 |
| HPP 82 | 21–25 | M | ODIMUNE | <20 | 834 | 36–40 | <1 | 317 | 4,20 | 55,56 | 425.99 |
| HPP 84 | 21–25 | M | ODIMUNE | <20 | 643 | 16–20 | <1 | 371 | 2,72 | 45,42 | NA |
| HPP 86 | 21–25 | F | ATROIZA | <20 | 1352 | 21–25 | 13 | 384 | 5,20 | 98,08 | 1057.08 |
| HPP 91 | 26–30 | F | ATROIZA | <20 | 843 | 21–25 | 14 | 482 | 2,46 | 31,14 | 884.47 |
| HPP 104 | 21–25 | M | ODIMUNE | <20 | 252 | 51–55 | <1 | 199 | 3,71 | 32,32 | NA |
| HPP 105 | 21–25 | F | ODIMUNE | <20 | 775 | 16–20 | 12 | 523 | 4,65 | 56,92 | 317.46 |
| HPP 109 | 21–25 | F | ATROIZA | <20 | 618 | 26–30 | 12 | 618 | 2,23 | 72,71 | NA |
| HPP 118 | 31–35 | M | ATROIZA | <20 | 339 | 16–20 | 1 | 165 | 5,30 | 48,47 | 592.30 |
| **Elite controllers** | | | | | | | | | | | |
| HPP 106 | 21–25 | F | ATROIZA | <20 | 782 | 31–35 | 11 | 612 | NA | 17,93 | NA |
| HPP 128 | 21–25 | M | ATROIZA | <20 | 579 | 11–15 | <1 | 479 | NA | 68,4 | NA |

*NA* Not available.

have identified to be crucial to ensure precise detection of *tat/rev* msRNA transcripts, were aligned to the complete HIV-1 genome alignment to assess primer-target sequence complementarity. To broaden the detection of multiple strains within a particular HIV-1 subtype, the primer binding regions with mutations present at sites extremely important for amplification were avoided by shifting these regions several nucleotides up or downstream. Melting temperature and distance between primer binding regions were adjusted to ensure that the modifications still met the LAMP primer specifications. Additionally, loop primers were generated using the PrimerExplorer V5 software, and one of these (e.g., Loop forward primer) was converted into a self-quenching probe with an internal FAM fluorophore bound to the closest thymidine from the 3′ end[63].

### RT-LAMP assay validation using in vitro *tat/rev* RNA transcripts

A gBlock comprising a spliced *tat/rev* sequence (subtype B or C), downstream of a T7 promoter sequence (generated by IDT) was used as a template for in vitro transcription using Hi-T7 RNA Polymerase and Ribonucleotide Solution Mix (both from NEB) in accordance with the manufacturer's instructions. The sequences of the subtype B and subtype C-specific *tat/rev* gBlocks are listed in Supplementary Table 3. To eliminate gBlock DNA, in vitro-transcribed *tat/rev* RNA samples were treated with RNase-free DNase I (NEB) and purified using the Monarch® RNA Cleanup Kit (NEB). Each purified sample of *tat/rev* RNA underwent quantification five times using a NanoDrop 2000 spectrophotometer (Thermo Fisher). The average values of the quantity for each sample, along with their respective lengths (509 bp for subtype B and 483 bp for subtype C *tat/rev* RNA), were employed to determine the number of RNA copies per microliter of the samples using NEBioCalculator (https://nebiocalculator.neb.com). Subsequently, the samples were serially diluted to achieve concentrations of 1000,

500, 250, 125, 50, 20, 10, 5, 1, and 0.1 copies of RNA per 5 µL. These diluted samples were utilized in validation experiments to assess the sensitivity of the RT-LAMP reactions. Reactions without the RTx enzyme were performed to assess the specificity of the RT-LAMP assay in an exclusively amplifying RNA template. The amplification was carried out in a CFX96 Touch Real-Time PCR Detection System thermocycler (Bio-Rad) following a thermal program of continuous 65 °C with fluorescence read every 30 s for 240 cycles, which corresponds to 120 min of amplification time. Complete details of the RT-LAMP reagents and conditions for reaction are listed in Supplementary Table 4. Positive reactions were identified by the presence of fluorescence curves exceeding the cutoff line and analyzed using.

### RT-LAMP-based detection of *tat/rev* msRNA in J-Lat 11.1 cells

Latent HIV-1-infected Jurkat cells (clone J-Lat 11.1), which harbor a fully integrated HIV-1 subtype B genome with a mutated *env* gene and a GFP reporter gene in place of *nef*, which requires multiple splicing[64], served as a surrogate for inducible reservoir cells. J-Lat 11.1 cells were kindly provided by Eric Verdin from the Buck Institute for Research on Aging. Single J-Lat 11.1 cells in the background of uninfected donor CD4+ T cells were used for the validation of RT-LAMP-based detection of *tat/rev* msRNA. J-Lat 11.1 cells were cultured in complete RPMI-1640 media, supplemented with 7% FBS and 100 µg/ml penicillin-streptomycin, and primary CD4+ T cells were cultured in RPMI-1640 media supplemented with 10% FBS and 100 µg/ml penicillin-streptomycin. J-Lat 11.1 cells were stimulated with 10 µM of phorbol 12 myristate 13-acetate (PMA) (Sigma) for 12 h and primary CD4+ T cells were stimulated with 100 ng/mL of PMA (Sigma) and 1 µg/mL of ionomycin (Sigma) for 12 h as mentioned elsewhere[35]. Activated uninfected donor-derived primary CD4+ T cells were washed twice in RPMI-1640 media supplemented with 3% FBS, counted using

Countess II (Thermo Fisher), and serially diluted to $4 \times 10^5$, $2 \times 10^5$, $1 \times 10^5$, $5 \times 10^4$, and $2.5 \times 10^4$ cells/mL in PBS. 5 μL of cell suspension from each dilution was mixed with 15 μL RT-LAMP master mix (Supplementary Table 4) and added to the PCR plate (Bio-Rad) in order to generate an increasing background of activated uninfected donor CD4+ T cells ($8 \times 10^4$, $4 \times 10^4$, $2 \times 10^4$, $1 \times 10^4$, and $5 \times 10^3$ cells/ 20 μL reaction, respectively). 5 μL of nuclease-free water was used instead of cell suspension to prepare PCR wells without any cell background. Activated J-Lat 11.1 cells were also washed twice in RPMI-1640 media supplemented with 3% FBS, and resuspended in PBS. A single GFP+ J-Lat 11.1 cell (marking a *tat/rev* msRNA+ cell) was sorted directly into each well of a 96-well PCR plate (Bio-Rad) containing 20 μL RT-LAMP master mix, without cells or with an increasing background of activated uninfected donor CD4+ T cells. The final reaction volume was maintained at 20 μL after the single-cell sorting as the sorted volume is negligible. RT-LAMP was conducted following a thermal program of continuous 65 °C with fluorescence read every 30 s for 180 cycles, which corresponds to 90 min of amplification time.

### Validating specificity for *tat/rev* HIV-1 msRNA by RT-LAMP

After 12 h stimulation of J-Lat 11.1 with PMA (see above)[35] either a single GFP+ or a single GFP− J-Lat 11.1 cell was sorted directly into each well of a 96-well PCR plate containing complete RT-LAMP master mix or the master mix without the reverse transcriptase (RT). RT-LAMP was conducted following the same conditions mentioned above. In order to compare the specificity of RT-LAMP that utilizes exon-spanning msRNA specific LAMP primers/probe with semi-nested RT-qPCR that utilizes non-exon-spanning primers/probe reported in previous study[35], a single GFP+ or GFP− J-Lat 11.1 cell was sorted directly into each well of a 96-well PCR plate and a semi-nested RT-qPCR was performed using previously reported reaction-mixes and thermal protocols[65].

Genomic DNA was isolated from PMA-stimulated J-Lat 11.1 cells using the phenol–chloroform-isoamyl alcohol isolation method and ethanol precipitation in the presence of glycogen as a carrier. Isolated DNA was treated with DNase-free Monarch® RNase A (New England Biolabs) to get rid of any RNA contamination. Total RNA was also isolated from PMA-stimulated J-Lat 11.1 cells using Trizol reagent (Sigma) according to the manufacturer's instructions. Isolated RNA was treated with RNase-free DNase I to get rid of any DNA contamination. 100 ng of DNase-treated RNA, 100 ng of RNase-treated DNA, and 100 copies of HIV-1 plasmid pNL4-3.Luc.R-E- were used as a template to perform semi-nested RT-qPCR following previously published protocol[65] and RT-LAMP on a CFX96 Touch Real-Time PCR Detection System thermocycler (Bio-Rad) following a thermal program mentioned above.

### SQuHIVLa validation for viral reservoir quantification

Next, we used SQuHIVLa to quantify the inducible reservoir in experimental samples that represent clinically observed reservoir size including 0.1, 1, 10, and 20 cells/million CD4+ T cells. To prepare these custom experimental samples, J-Lat 11.1 cells and CD4+ T cells derived from uninfected donor PBMCs were cultured and stimulated following the same procedure mentioned in the previous section. After stimulation, CD4+ T cells were washed, counted using an automated cell counter (Countess II, Thermo Fisher) and four tubes containing 10 million CD4+ T cells resuspended in 10 mL of RPMI-1640 media supplemented with 3% FBS (1 million CD4+ cells/mL) were prepared. A range of 1, 10, 100, or 200 GFP + J-Lat 11.1 cells were sorted into each tube generating custom test samples representative of clinical samples with an inducible HIV-1 subtype B reservoir of 0.1, 1, 10, 20 cells/million CD4+ T cells. The cells in each tube were then washed, resuspended in phosphate-buffered saline (PBS), and then serially diluted to $4 \times 10^6$ cells/ml, $2 \times 10^6$ cells/ml, $1 \times 10^6$ cells/ml, and $5 \times 10^5$ cells/ml in PBS. From each dilution, 5 μL of the cell suspension was distributed to 22–24 wells of a 96-well plate containing 15 μL RT-LAMP master mix (Supplementary Table 4) corresponding to 20,000, 5000, 1250, and 313 cells per well, similar to the previously reported study by Procopio et al. [35] although the number of cells per well and the amplification

procedure is different. After the RT-LAMP reaction, the positive wells at each dilution were scored, and the maximum likelihood method was used to determine the frequency of cells expressing *tat/rev* msRNA using the IUPMStats v1.0 online software[66].

Additional SQuHIVLa validation experiments were performed using primary CD4+ T cells from people living with HIV (PLWH). Isolated CD4+ T cells were resuspended in culture RPMI-1640 media supplemented with 10% FBS and 100 μg/ml penicillin-streptomycin and rested for 5 h at 37 °C in a humidified, 5% $CO_2$ incubator. CD4+ T cells were then stimulated with 100 ng/mL of PMA and 1 μg/mL of ionomycin (both from Sigma) for 12 h as described previously[35]. Activated CD4+ T cells were washed, counted, serially diluted in PBS, and distributed to a 96-well PCR plate containing 15 μL RT-LAMP master mix (Supplementary Table 4) following the same procedure mentioned above. RT-LAMP was carried out following a thermal program; incubation at 45 °C for 60 min followed by continuous 65 °C with fluorescence read every 30 s for 180 cycles, which corresponds to 90 min of amplification time (unless mentioned otherwise). After the RT-LAMP reaction, the frequency of cells expressing *tat/rev* msRNA was quantified using IUPMStats v1.0 online software[66] as mentioned above.

When a limited number of PBMCs were available, activated CD4+ T cells were serially diluted to $2 \times 10^6$ cells/ml, $1 \times 10^6$ cells/ml, $2.5 \times 10^5$ cells/ml, and $6.3 \times 10^4$ cells/ml in PBS and, 5 μL of the cell suspension was distributed to 22–24 wells of a 96-well plate corresponding to 10,000, 5000, 1250, and 313 cells per well.

### *tat/rev* Induced Limiting Dilution Assay (TILDA)

CD4+ T cells obtained from PLWH were stimulated for 12 h with PMA and ionomycin and serially diluted in a limiting fashion as mentioned previously[35]. Semi-nested PCR was performed using the reaction mix and thermal protocols reported by Procopio et al. [35] and Lungu et al. [65]. TILDA Primers were reported in a previous study[35].

### Intact Proviral DNA Assay (IPDA)

Intact proviral DNA levels in CD4+ T cells were evaluated using lysed extracts, with nucleic acid isolation from at least $2 \times 10^6$ CD4+ T cells via the PCI extraction method. Duplex digital PCR (dPCR) on an Absolute Q digital PCR platform (Thermo Fisher) targeted the packaging signal (Ψ) and Envelope gene (Env) following an adapted IPDA potocol[18,67,68]. Briefly, the PCR reaction mix includes 1.8 μL Absolute Q DNA Digital PCR Master Mix 5X (Thermo Fisher Scientific), 1 μL of the packaging signal (Ψ) target FAM-probe (10×), 1 μL of the Env target HEX-probe (10×), and 700 ng of genomic DNA (gDNA). This mixture was diluted in nuclease-free $H_2O$ to achieve a final reaction volume of 9 μL. The frequency of CD4+ T cells was determined by measuring the cellular gene ribonuclease P/MRP subunit p30 (RPP30) in a duplicate well. The duplex RPP30 dPCR was arranged in a similar manner, employing RPP30 primer and probe sets with a 7 ng genomic DNA (gDNA) input[18].

### Quantification of total HIV-1 DNA measured by ddPCR

Total HIV-1 DNA was measured by droplet digital PCR (ddPCR) (Bio-Rad) as previously described[69]. Genomic DNA was isolated from CD4+ T cells using DNeasy Blood & Tissue Kits (QIAGEN). Quantification of the RPP30 gene was used to determine host cell concentration. Amplification was performed with primers and probes covering HIV-1 5′ LTR-gag HXB2 coordinates 684–810[69]. Thermocycling conditions for ddPCR were: 95 °C for 10 min, 45 cycles of 94 °C for 30 s and 60 °C for 1 min, 72 °C for 1 min. Thereafter droplets from each sample were analyzed on the Bio-Rad QX200 Droplet Reader and data were analyzed using QuantaSoft software (Bio-Rad).

### Statistics and reproducibility

All graphs were generated and statistical analyses were performed using GraphPad Prism version 8.0.2 for Windows (GraphPad Software, San Diego, California USA, www.graphpad.com). In order to calculate the lowest amount of *tat/rev* msRNA in a sample that can be consistently

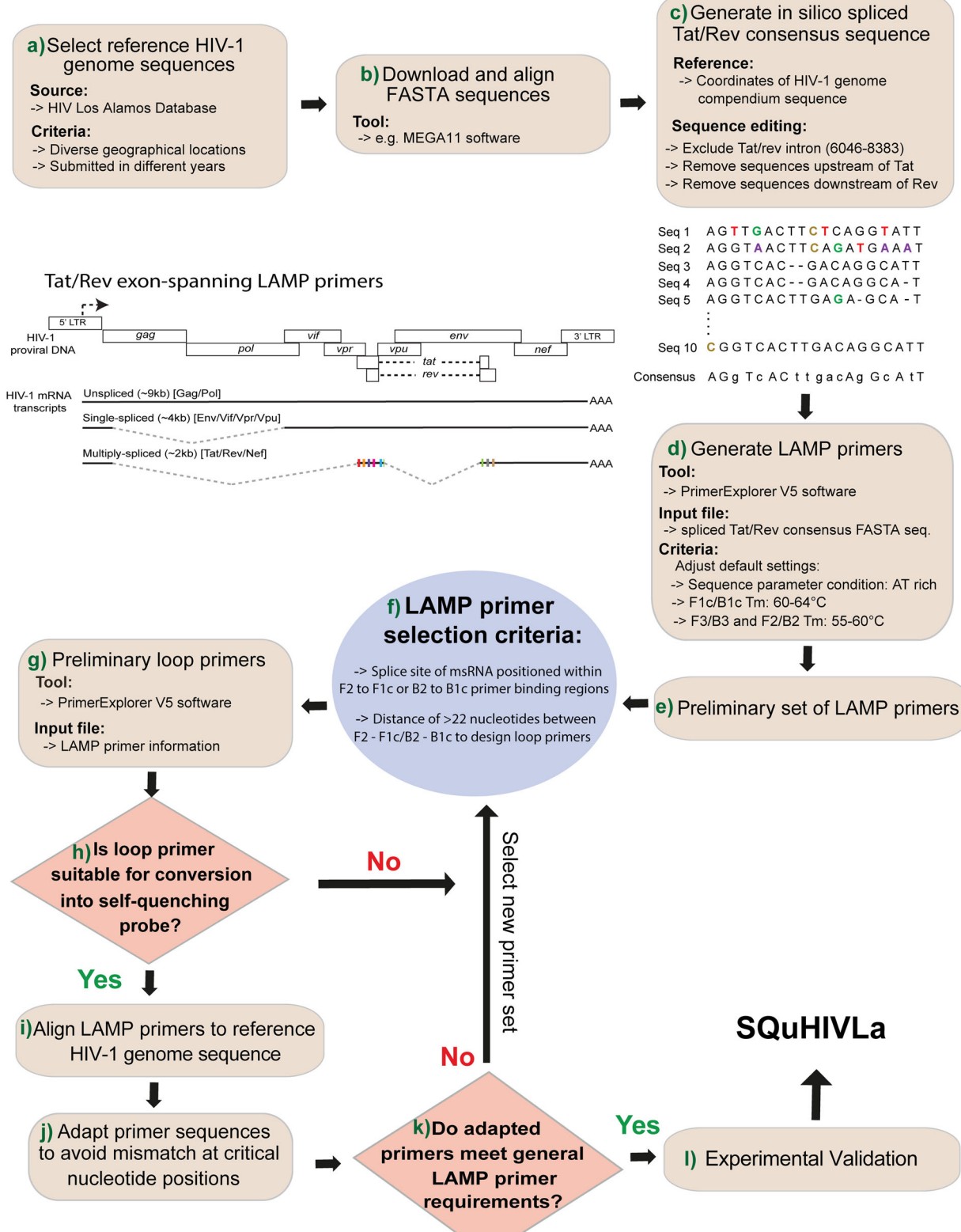

**Fig. 1 | Flowchart of sequential steps in designing *tat/rev* msRNA LAMP primers/ probe set.** This flowchart outlines the steps for designing LAMP primers/probes for tat/rev msRNA detection. First, ten complete HIV-1 subtype B genome sequences are retrieved (step a) and aligned (step b) to create a consensus sequence (step c). Using this consensus, preliminary LAMP primer sets are generated with Primer-Explorer V5 software (steps d, e). Primers with exon-spanning binding regions are selected to avoid intron-containing genomic DNA amplification (step f). Loop primers are generated (step g) and selected for conversion into self-quenching probes (step h). The selected primers are aligned with HIV-1 sequences to ensure complementarity (step i) and modified if necessary to minimize the impact of mutation hotspots (step j). Primers meeting all criteria are chosen (step k), and various experimental validations are conducted before the primer set is finalized for use in SQuHIVLa (step l). Detailed methods are provided in Supplementary Note 1.

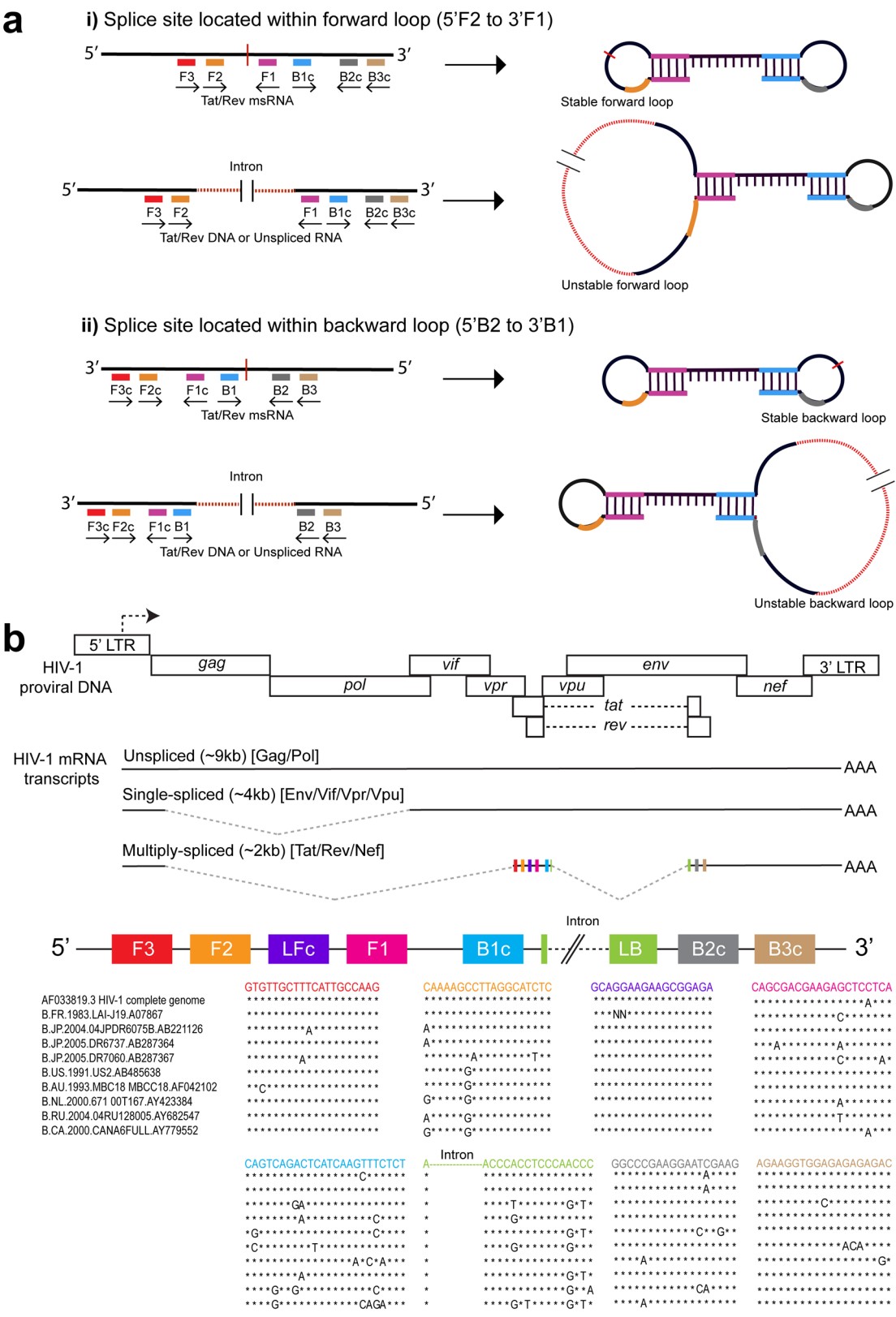

**Fig. 2 | Schematic overview of SQuHIVLa and primer binding sites specific for HIV-1 subtype B. a** Schematic overview of the primer binding regions where the splice site is located within the forward or backward loop. Primers bound to msRNA results in a stable forward and backward loop formation. However, the presence of intron (in case of *tat/rev* DNA or usRNA transcript) leads to an unstable forward or backward loop formation, depending on the position of the intron within 5'F2 to 3' F1 or 5'B2 to 3'B1 respectively. **b** Schematic overview of HIV-1– genome and transcripts produced during the HIV-1 replication cycle and alignment of the different LAMP primer binding regions with Ten HIV-1 subtype B viral sequences. The solid line represents the exons and the dotted lines represent the intron that is excised to generate mature HIV-1 mRNA transcripts. Exons of a multiply spliced RNA transcripts have colored rectangles that represent approximate LAMP primer binding sites. Similar to the alignment, different colors corresponds to different regions of the primers. After alignment, the "*" symbol is used to denote identical nucleotides; otherwise, the actual nucleotide symbol (A, T, C, or G) is used.

detected with 95% probability (LOD-95%), probit analysis was used. The Probit was calculated by the Excel function $[5 + \text{NORMSINV}(P)]$, where $P$ was the HIT rate (proportion of positive reactions) and 6.64 probit value (1.64 for 95% limit, +5 for probit scale) was used to extrapolate LOD-95% RNA copies. The effect of background cells on RT-LAMP amplification time was determined using Ordinary One-way ANOVA followed by Dunnett's multiple comparisons test. The coefficient of variation (CV) was calculated to determine the reproducibility of SQuHIVLa. Accuracy percentage (AP) was calculated using the ABS function in Excel. Normality of the HIV-1 reservoir sizes measured by SQuHIVLa, TILDA, IPDA, and total HIV-1 DNA was tested with the Shapiro–Wilk test. Correlations were performed using the Pearson test when data passed the normality test, otherwise, Spearman's rank correlation test was used. Band-Altman calculations were performed to test the agreement between reservoir size measured by SQuHIVLa and TILDA. Inducible HIV-1 subtype C reservoir size quantified from PLWH, grouped based on biological sex, were compared utilizing a two-tailed unpaired $t$-test. Experiments that used J-Lats 11.1 cells were performed at least three times, or as indicated in the figure legends. For experiments using primary CD4+ T cells obtained from human donors, sample sizes were indicated in the figure legends. The sample size was determined based on the minimum requirement to perform statistical tests and the availability of materials. All the experimental schematics were generated using BioRender.com.

### Reporting summary

Further information on research design is available in the Nature Portfolio Reporting Summary linked to this article.

## Results

### A LAMP primer/probe set designed for exclusive detection of HIV-1 *tat/rev* msRNA

The successful design of a LAMP primer/probe set for specific LAMP detection of HIV-1 *tat/rev* msRNA, depended on a set of guidelines as depicted in Fig. 1. The most critical criterion to ensure precise detection of *tat/rev* msRNA, excluding intron-containing *tat/rev* HIV-1 genomic DNA, was that the primer binding sites are exon-spanning. Using HIV-1 subtype B as a reference, we designed primer/probe sets as depicted in the flowchart (Fig. 1). We retrieved (Fig. 1 step a) and aligned (Fig. 1 step b) ten complete genome sequences of HIV-1 subtype B from different years (1983–2005) and various geographical locations to generate an *in silico*-spliced *tat/revv* HIV-1 DNA consensus sequence (Fig. 1 step c). Using the PrimerExplorer V5 software and the consensus *tat/rev* HIV-1 DNA FASTA file, we generated a list of preliminary sets of LAMP primers (Fig. 1 steps d and e) from which we selected primer sets where either the F2 binding region was within nucleotide position 125–240 or the B2 binding region was within nucleotide position 190–305, to ensure that the splice site (215th nucleotide) was located within the forward loop (5′F2 to 3′F1) or backward loop (5′B2 to 3′B1) (Fig. 1 step f). This criterion ensures that unstable loop formation resulting from primer binding to intron-containing *tat/rev* DNA, would inhibit isothermal amplification, thereby resulting in the exclusive detection of *tat/rev* msRNA (Fig. 2a). Additionally, we generated a set of loop primers using the PrimerExplorer V5 software (Fig. 1, step g) and selected the loop primer pair where at least one of them is suitable for the conversion into a self-quenching probe (Fig. 1 step h, Methods). Specific details are provided in the Supplementary Note 1. The selected set of LAMP and loop primers were aligned to the ten HIV-1 genome sequences to assess primer-target sequence complementarity (Fig. 1 step i and Fig. 2b). Given HIV-1's high genetic diversity, it is impossible to ensure that the primer binding regions in patient-specific viral RNA are devoid of mutation hotspots. However, we strived to ensure that the mutation hotspots are located at positions such as the 5′ ends of F3, B3, F2, B2, and 3′ ends of F1c, B1c primer binding regions (Fig. 1 step j and Fig. 2b). A set of primers with mismatches at these positions, whilst still meeting all the LAMP primer criteria such as melting temperature and distance between primer binding sites, would have the least impact on amplification (Fig. 1 step k and "Methods" section) hence selected

as the final primer set. The loop forward primer (LF) was converted into a self-quenching probe by adding an internal FAM fluorophore to enable sequence-specific, real-time detection and quantification of LAMP amplicons ("Methods" section, Supplementary Note 1). Additionally, a variety of experimental validation approaches, performed in this work, were applied to the final primer set before it was selected for use in SQuHIVLa (Fig. 1 step l). The designed primers and probe sequences are provided in Supplementary Table 2.

### Highly sensitive and specific detection of *tat/rev* msRNA by RT-LAMP

To assess the sensitivity of RT-LAMP in detecting *tat/revv* msRNA, we used serial dilutions of msRNA, in vitro transcribed from an intron-free *tat/rev* DNA template, driven by a T7 promoter (Fig. 3a). As few as 50 copies of RNA were detected within 30.46 min (±7.21 min) using RT-LAMP in >97% of all reactions, with a Limit of Detection-95% at 31 copies (Fig. 3b, d, e). At lower RNA copy numbers, RT-LAMP exhibited reduced efficiency, resulting in a longer time to results and increased variability between reactions, though all positive amplifications were achieved within 90 min (Fig. 3c). In experiments to assess the specificity of Bst 2.0 polymerase enzyme for *tat/rev* msRNA, we performed LAMP reactions with the required Bst 2.0 polymerase enzyme in the absence of the reverse transcriptase enzyme, RTx (Methods). Minimal and inconsistent amplification was observed at high copies of RNA (≥250 copies), suggesting some basal reverse transcriptase activity of the Bst 2.0 polymerase, which manufacturer validation experiments have also shown. However, the RT enzyme is necessary for robust and consistent amplification of RNA templates (Supplementary Fig. 1A, B).

Subsequently, to determine the sensitivity of RT-LAMP in detecting *tat/rev* msRNA directly from cells, we used J-Lat 11.1 cells, a Jurkat cell line harboring a latent but inducible full-length HIV-1 genome that expresses green fluorescence protein (GFP), as a product of msRNA transcripts (Supplementary Fig. 1C), upon activation. We sorted single GFP+ cells (Supplementary Fig. 1D) into RT-LAMP reaction plates in the presence or absence of PMA-stimulated uninfected donor CD4+ T cells (Fig. 3f). In the absence of CD4+ T cell background, a single GFP+ cell was detected in 95.83% of all RT-LAMP reactions (Fig. 3g), decreasing to 83.33% positivity in a background of $2 \times 10^4$ cells, however, the presence of cell background did not significantly impact the amplification time of RT-LAMP (mean amplification time ranging from 31.26 min for no cell background to 42.63 min in a background of $8 \times 10^4$ cells) (Fig. 3h, i).

Importantly, in an additional series of validation experiments, we made a crucial observation demonstrating the selective amplification of *tat/rev* msRNA and the exclusion of intron-containing *tat/rev* genomic DNA when employing RT-LAMP (Fig. 4a, b). This specificity was in stark contrast to the results obtained using semi-nested PCR with non-exon-spanning primers/probe, which detected both the msRNA and genomic DNA, as illustrated in Supplementary Fig. 2. Minimal to no amplification was observed when RT-LAMP reactions were performed without the reverse transcriptase enzyme (Fig. 4a, Supplementary Fig. 3A) as well as in a range of validation samples including; GFP− J-Lat 11.1 cells that do not express msRNA (Fig. 4a, Supplementary Fig. 3B); DNase-treated J-Lat 11.1 RNA and RNase-treated J-Lat 11.1 DNA or using HIV-1 plasmid as template (Fig. 4b). Overall, we demonstrate the rapid, sensitive and specific detection of cells expressing *tat/rev* msRNA by RT-LAMP and sought to use this assay for quantifying the inducible HIV-1 reservoir, hereafter referred to as SQuHIVLa (Specific Quantification of Inducible HIV-1 reservoir by RT-LAMP).

### Accurate and reproducible quantification of the frequency of *tat/rev* msRNA-expressing cells using SQuHIVLa

In the next set of experiments, we sought to assess the reproducibility of SQuHIVLa in quantifying the frequency of *tat/rev* msRNA-expressing cells, a surrogate measure for the inducible viral reservoir. To model HIV-1 reservoir samples, we generated a panel of four test samples comprising

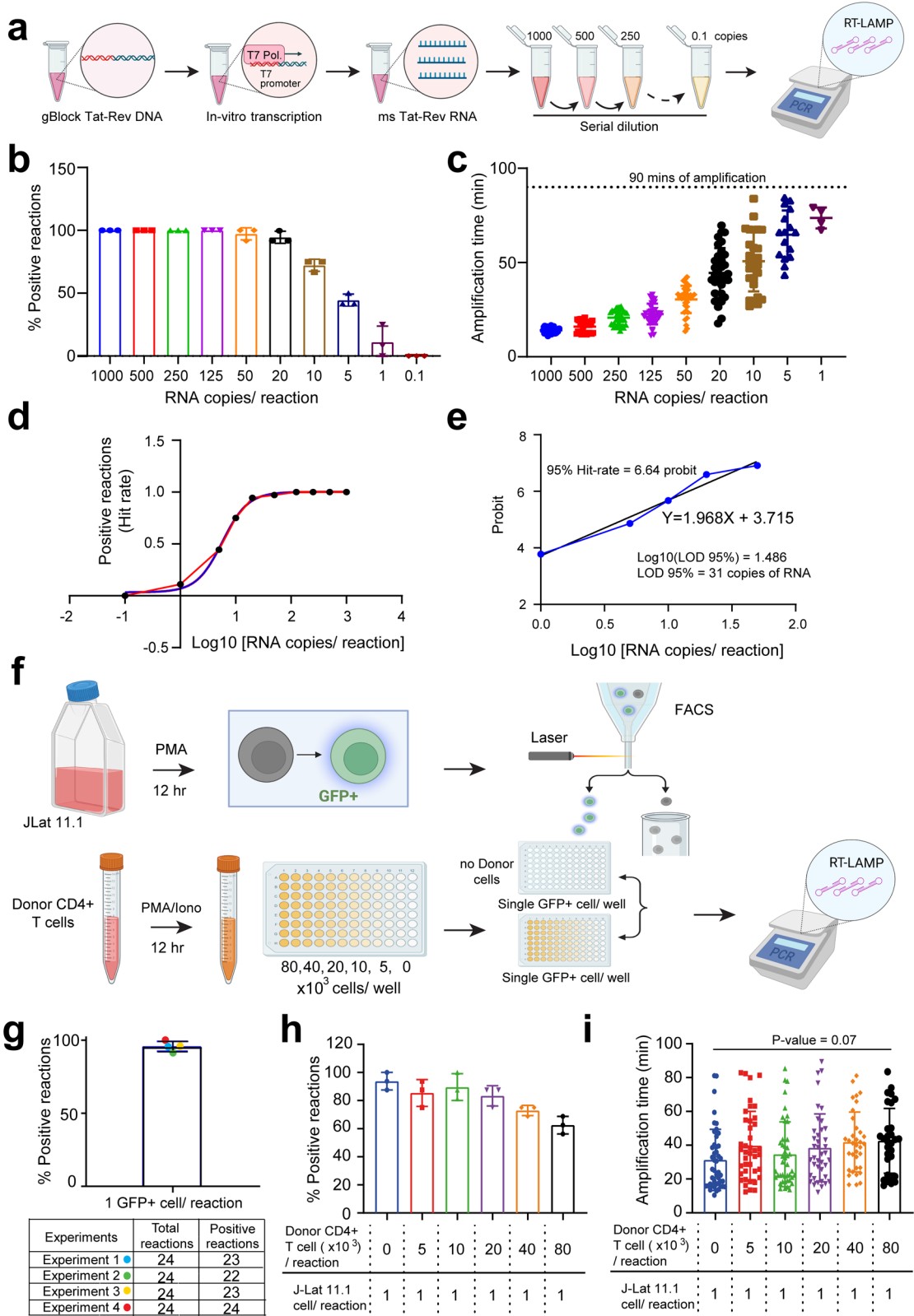

activated GFP+ J-Lat 11.1 cells (0.1, 1, 10 and 20 cells) sorted into a suspension of one million CD4+ T cells. Cells from each test sample were then distributed into RT-LAMP reaction plates in a limiting dilution format, which allows quantification by maximum likelihood statistics (Fig.4c, "Methods" section). The observed and predicted viral reservoir size measurement were highly precise at frequencies greater than 20 GFP+ cells/

million CD4+ T cells with a coefficient of variation (CV) of 6.61%. The CV increased to 60.61% at 0.1 cell GFP+ cells/million CD4+ T cells (Fig. 4d). Furthermore, the observed and predicted viral reservoir measurements correlated significantly having a good accuracy with mean accuracy percentage (AP) of 85% for reservoir size greater than 1 GFP+ cell/million CD4+ T cells (Fig. 4e).

**Fig. 3 | Sensitivity of RT-LAMP used in SQuHIVLa to detect msRNA.**
**a** Experimental outline to generate synthetic msRNA using T7 polymerase mediated by in vitro transcription. Synthetic msRNA is serially diluted to achieve samples containing a range of target RNA copies used to perform RT-LAMP. **b** Percentage of positive RT-LAMP reactions and **c** amplification time required for a range of RNA copies are plotted. Data are presented as mean ± SD of three independent in vitro transcription experiments followed by serial dilution and RT-LAMP. Twelve reactions were performed per condition for each independent experiments. Dotted line represents 90 min of isothermal amplification. **d** A non-linear regression analysis is performed using the proportion of positive RT-LAMP reactions (HIT rate) (Y-axis) corresponding to Log10 values of RNA copies used as a template (X-axis). **e** Probit analysis is performed to determine LOD-95% using probit values calculated from HIT rate (Y-axis) corresponding to Log10 values of RNA copies (Y-axis).
**f** Experimental outline to sort single GFP+ J-Lat 11.1 cell in each well of 96-well PCR

plate containing either no cells or different number of stimulated healthy donor CD4+ T cells to determine the sensitivity of RT-LAMP. **g** Percentage of RT-LAMP positive reactions when a single GFP+ J-Lat 11.1 cell/reaction is probed in the absence of cellular background. Data are presented as mean ± SD of four independent experiments (24 reactions per experiment) and each experiment is represented with different colored dots. **h** Percentages of RT-LAMP positive reactions when a single GFP+ J-Lat 11.1 cell/reaction is probed in the presence of an increasing background of uninfected donor CD4+ T cells and **i** their corresponding RT-LAMP amplification time. Data are presented as mean ± SD of three independent experiments. Sixteen reactions were performed per condition for each independent experiments. Ordinary one-way ANOVA followed by Dunnett's multiple comparison test is performed to analyze the variation of amplification times among different conditions and statistical significance is determined by $p < 0.05$.

To further validate SQuHIVLa, we applied the assay to quantify the frequency of msRNA+ cells as a surrogate measure for an inducible viral reservoir in primary CD4+ T cells obtained from people with HIV-1 subtype B (PLWHB) on suppressive antiretroviral therapy. Following 12-h PMA/ionomycin stimulation, which enhances transcriptional activation of inducible proviruses, whole CD4+ T cells were distributed in a limiting dilution format and *tat/rev* msRNA transcripts were amplified using RT-LAMP (Fig. 4f). The frequency of cells expressing *tat/rev* msRNA was then determined by maximum likelihood estimation. We observed suboptimal *tat/rev* msRNA amplification in primary CD4+ T cells, and this was addressed by optimizing the sample incubation time at 45 °C before isothermal amplification at 65 °C. This adjustment facilitated more efficient RNA release. Importantly, the viral reservoir size did not exhibit an increase in three independent donors with incubation times exceeding one hour at 45 °C (Fig. 4g). This optimization step was subsequently incorporated into the SQuHIVLa protocol, along with 90 min of isothermal amplification, as all positive signals were consistently detected within the initial 90 min of amplification (Supplementary Fig. 4). The mean inter-assay CV, 9.58% (Range: 7.2%–12.23%) was determined from three independent SQuHIVLa experiments using CD4+ T cells from three PLWHB demonstrating high reproducibility of SQuHIVLa in quantifying the inducible HIV-1 reservoir (Fig. 4h).

## Correlation of reservoir size measured by SQuHIVLA with several assays
In this section, we compared the quantification of the inducible msRNA+ reservoir size using SQuHIVLa and TILDA, a well-established assay for assessing the frequency of msRNA+ cells. Specifically, we measured the inducible *tat/rev* msRNA+ reservoir size for 4 PLWHB using both SQuHIVLa and the TILDA protocol reported by Procopio et al. (TILDA_Procopio)[35] and for 9 PLWHB by SQuHIVLa and the TILDA protocol reported by Lungu et al. (TILDA_Lungu)[65]. These two TILDA versions have been previously documented for their excellent reproducibility, demonstrating low inter-laboratory variability[65]. The correlation between SQuHIVLa and TILDA_Lungu ($r = 0.62$, $P = 0.0857$, Fig. 5c) was stronger than with TILDA_Procopio ($r = 0.2$, $p = 0.9167$, Fig. 5a) as indicated by Spearman's rank correlation coefficient ($r$), though statistical significance was not achieved for either comparison, possibly due to the limited sample size ($n = 4$) in the SQuHIVLa vs TILDA_Procopio and ($n = 9$) SQuHIVLa vs TILDA_Lungu analyses. However, Band-Altman plots demonstrated strong agreement of SQuHIVLa values with both TILDA_Procopio (mean bias = 0.058, 95% limits of agreement (LOA) = −0.48 to 0.59, Fig. 5b) and TILDA_Lungu (mean bias = 0.051, 95% LOA = −0.31 to 0.41, Fig. 5d). Values measured by SQuHIVLa and TILDA_Lungu ($n = 9$) also exhibited moderate correlation, with a Lin's concordance correlation coefficient (LCCC) of 0.62 and coefficient of bias (Cb) of 0.9714. Lin's concordance correlation was not calculated for the comparison between SQuHIVLa and TILDA_Procopio due to the limited sample size ($n = 4$).

Furthermore, we explored the correlation between values obtained from primary CD4+ T cells of 12 PLWHB using SQuHIVLa and their

corresponding total HIV-1 DNA copies and intact HIV-1 DNA copies per million CD4+ T cells measured by IPDA. The values of SQuHIVLa showed a significant positive correlation with intact HIV-1 DNA copies ($r = 0.58$, $p = 0.049$), but not with total HIV-1 DNA copies ($r = 0.21$, $p = 0.5199$) (Fig. 5f, g).

## Adaptation of SQuHIVLa for non-B HIV-1 Subtypes
To determine the feasibility of using one primer-probe set for different HIV-1 subtypes, we aligned the subtype B *tat/rev* msRNA LAMP primer-probe sequences to genome sequences of HIV-1 subtypes C and A. A heatmap of the mismatch score, corresponding to the relative quantity of matching nucleotides for each primer binding region, revealed poor compatibility of the subtype B LAMP primer-probe sets for detection of subtypes C and A *tat/rev* msRNA (Supplementary Fig. 5). Several mismatches were present at sites critical for the LAMP reaction. Therefore, to specifically detect *tat/rev* msRNA from people with HIV-1 subtype C, globally the most prevalent subtype, we designed a new set of LAMP primers and probes (Fig. 6a) following the aforementioned guidelines (Fig. 1, "Methods" section and Supplementary Note 1) and fulfilling the requirements to ensure specific amplification of subtype C *tat/rev* msRNA. The new primer-probe set demonstrated similar sensitivity in detecting *in vitro* HIV-1 C *tat/rev* msRNA transcripts compared to the HIV-1 subtype B primer/probe set. In all experiments performed, 50 copies of *tat/rev* HIV-1 subtype C msRNA were detected in >97% reactions (Fig. 6b) with a mean amplification time of 38.19 min (±7.47 min) (Fig. 6c). All positive amplifications were achieved within 90 min (Fig. 6c) and the LOD-95% was determined to be 45 copies (Fig. 6d, e), consistent with the observations for Subtype B *tat/rev* msRNA. To assess the cross-reactivity of the primers/probe, inducible reservoirs were quantified in three individuals with HIV-1 subtype B (PLWHB) and three with subtype C (PLWHC). This analysis involved the use of both subtype-specific (subtype B and subtype C-specific primers/probe for PLWHB and PLWHC, respectively) and non-specific (subtype C and subtype B-specific primers/probe for PLWHB and PLWHC, respectively) LAMP primers/probe sets. Suboptimal reservoir quantification was observed when subtype non-specific primers/probe were utilized (Fig. 6f), highlighting the importance of designing subtype-specific primers/probe sets for optimal reservoir quantitation.

We then assessed the performance of SQuHIVLa across a panel of clinical samples from 19 PLWHC on fully suppressive ART (17 chronically infected and 2 elite controllers). *tat/rev* msRNA was detectable in all samples and the frequency of cells expressing *tat/rev* msRNA ranged from 10.12 to 98.08 cells per million CD4+ T cells (median = 48.47, IQR 32.32 −68.40) (Fig. 6g). Furthermore, we explored the correlation between values obtained from primary CD4+ T cells of 12 PLWHC using SQuHIVLa and their corresponding total HIV-1 DNA copies measured by digital droplet PCR. SQuHIVLa values did not show a statistically significant positive correlation with total HIV-1 DNA copies (Pearson $r = 0.2083$, $P$-value = 0.5160). Interestingly, in the chronically infected PLWHC, the frequency of cells expressing *tat/rev* msRNA was notably higher in females ($n = 10$) than in males ($n = 7$) ($P = 0.0202$) (Fig. 6i). Univariate correlation analyses did not

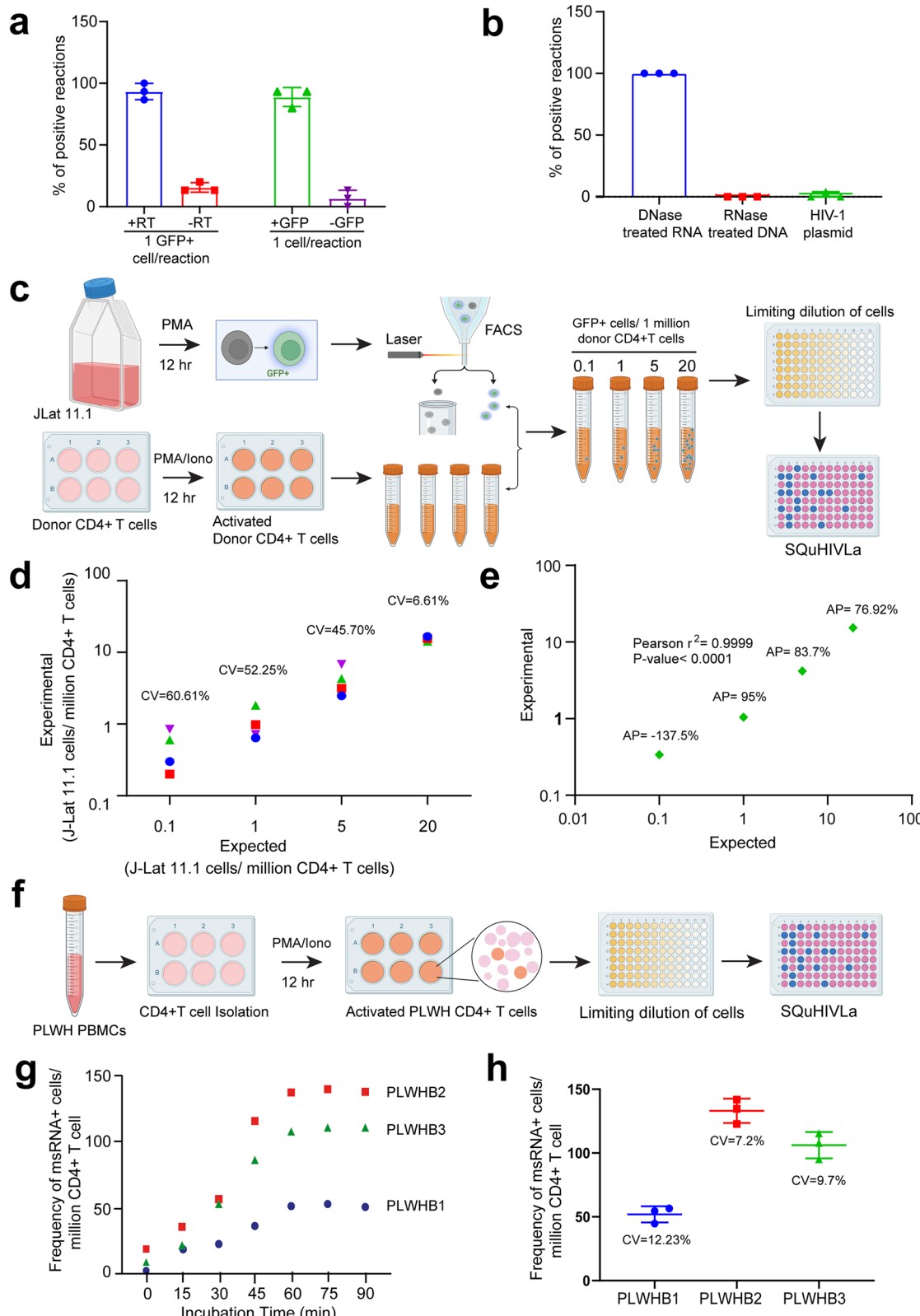

reveal significant association between the frequency of cells expressing *tat/rev* msRNA and CD4+ T cell counts or pre-ART plasma HIV-1 RNA or duration of infection before ART initiation (Supplementary Fig. 6) Overall, these findings demonstrate that SQuHIVLa is highly adaptable and has great potential for application in studies involving people with HIV-1, with various clinical characteristics.

## Discussion

Previous and emerging studies have reported that *tat/rev* msRNA levels correlate with plasma HIV-1 RNA[42] and that expression of cell-associated msRNA correlates with and may be a potential biomarker predictive of cellular viral rebound upon ART interruption[70–72]. A strong correlation between supernatant HIV-1 RNA and msRNA upon ex vivo latency

**Fig. 4 | Quantification of inducible HIV-1 reservoir using SQuHIVLa.**
**a** Percentages of RT-LAMP positive reactions when a single GFP+ J-Lat 11.1 cell/reaction was probed with or without reverse transcriptase enzyme included in the RT-LAMP reaction, and when either a single GFP+ or GFP− J-Lat 11.1 cell/reaction was probed. **b** Percentages of positive reactions when a DNase-treated RNA and RNase-treated DNA sample isolated from PMA-stimulated J-Lat 11.1 cells, or when pNL4.3 E-R- plasmid were used as template for RT-LAMP. **c** Experimental outline of preparing custom samples with an inducible HIV-1 reservoir of 0.1, 1, 5, 20 GFP+ cells/millions of CD4+ T cells using stimulated J-Lat 11.1 cells and uninfected donor CD4+ T cells. These custom samples were used to quantify the reservoir size using the SQuHIVLa assay. **d** The inter-assay coefficient of variation (CV) is determined from four independent experiments for all four samples representing different inducible reservoir size (0.1, 1, 5, 20 GFP+ cells/million CD4+ T cells). Four different colored symbols represent data from four individual experiments. **e** The

Pearson correlation coefficient ($r^2$) is determined between expected and experimental inducible reservoir size and statistical significance is determined by $p < 0.05$. The accuracy percentage (AP) was calculated with Excel using ABS function. **f** A brief experimental outline of the application of SQuHIVLa to quantify inducible HIV-1 subtype B reservoir using PMA/ionomycin-activated CD4+ T cells in limited dilution format followed by maximum likelihood calculation. **g** The frequency of msRNA+ cells as a surrogate measure for inducible HIV-1 reservoir was quantified for three different people living with HIV-subtype B (PLWHB) using a SQuHIVLa protocol adapted to include an incubation at 45 °C prior to RT-LAMP in order to determine the optimal incubation time for primary samples. Three different colored symbols represent data of three PLWHB. **h** The inter-assay CV is determined from three independent experiments for three PLWHB. Data are presented as mean ± SD and three different colored symbols represent three PLWHB.

**Fig. 5 | Correlation Analyses of SQuHIVLa with Various Reservoir Quantification Assays.**
**a** Spearman ($r$) correlation plot and **b** Bland–Altman plot depicting log-transformed inducible reservoir size measurements ($n = 4$) by SQuHIVLa and TILDA_Procopio. The green dotted line indicates mean bias, and blue dotted lines represent 95% limits of agreement. **c** Spearman ($r$) correlation plot and **d** Bland–Altman plot illustrating log-transformed inducible reservoir size measurements ($n = 9$) by SQuHIVLa and TILDA_Lungu. Blue dotted line signifies mean bias, and green dotted lines represent 95% limits of agreement. **e** Lin's Concordance Correlation Coefficient (LCCC) plot. Green triangles depict inducible reservoir size quantified by SQuHIVLa and TILDA_Lungu. Blue diagonal line represents the best-fit line. Coefficient of bias (Cb) serves as a measure of accuracy. **f** Pearson ($r$) correlation between SQuHIVLa and total HIV-1 DNA copies and **g** intact HIV-1 DNA copies for 12 PLWHB. Statistical significance determined by $p < 0.05$.

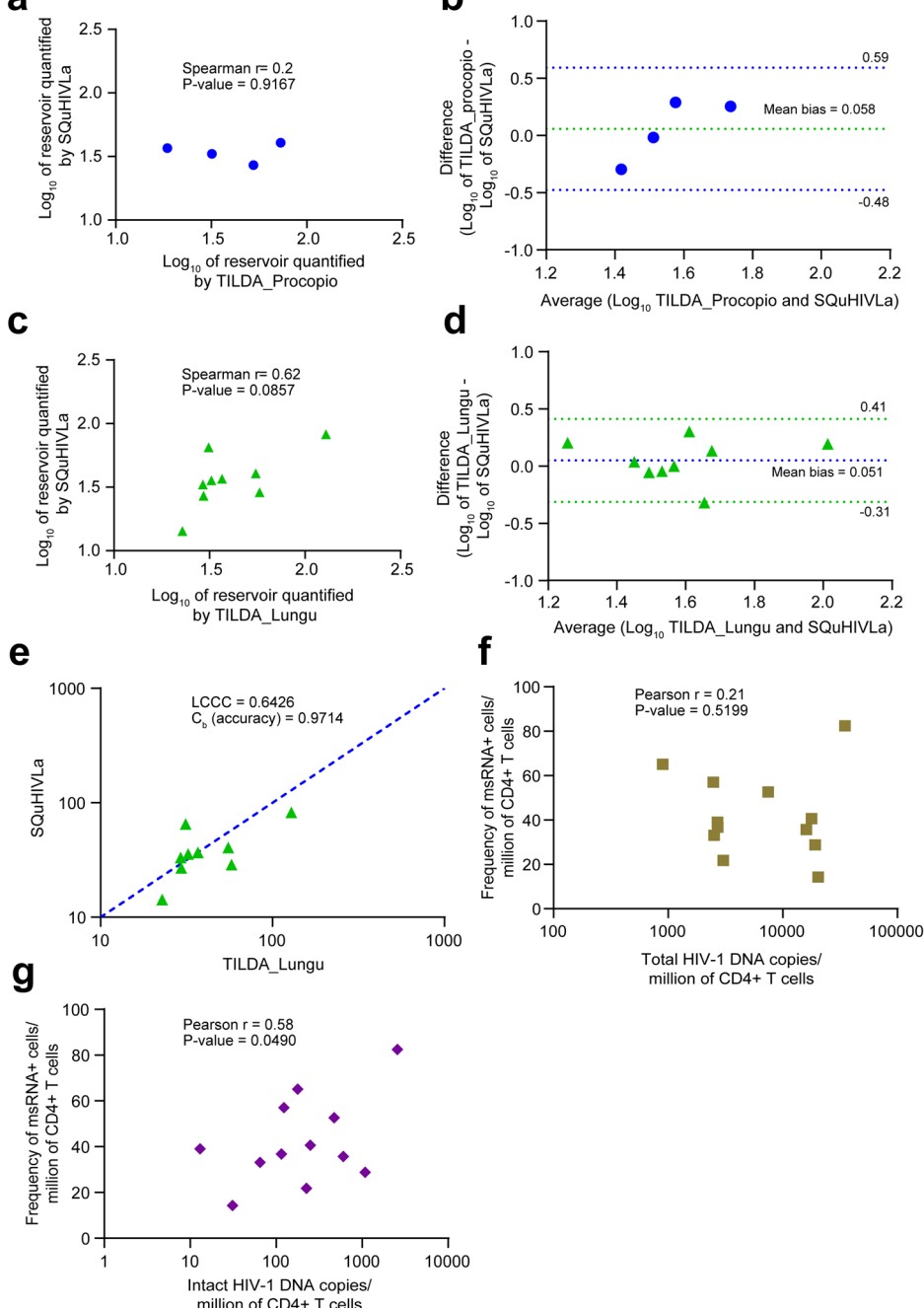

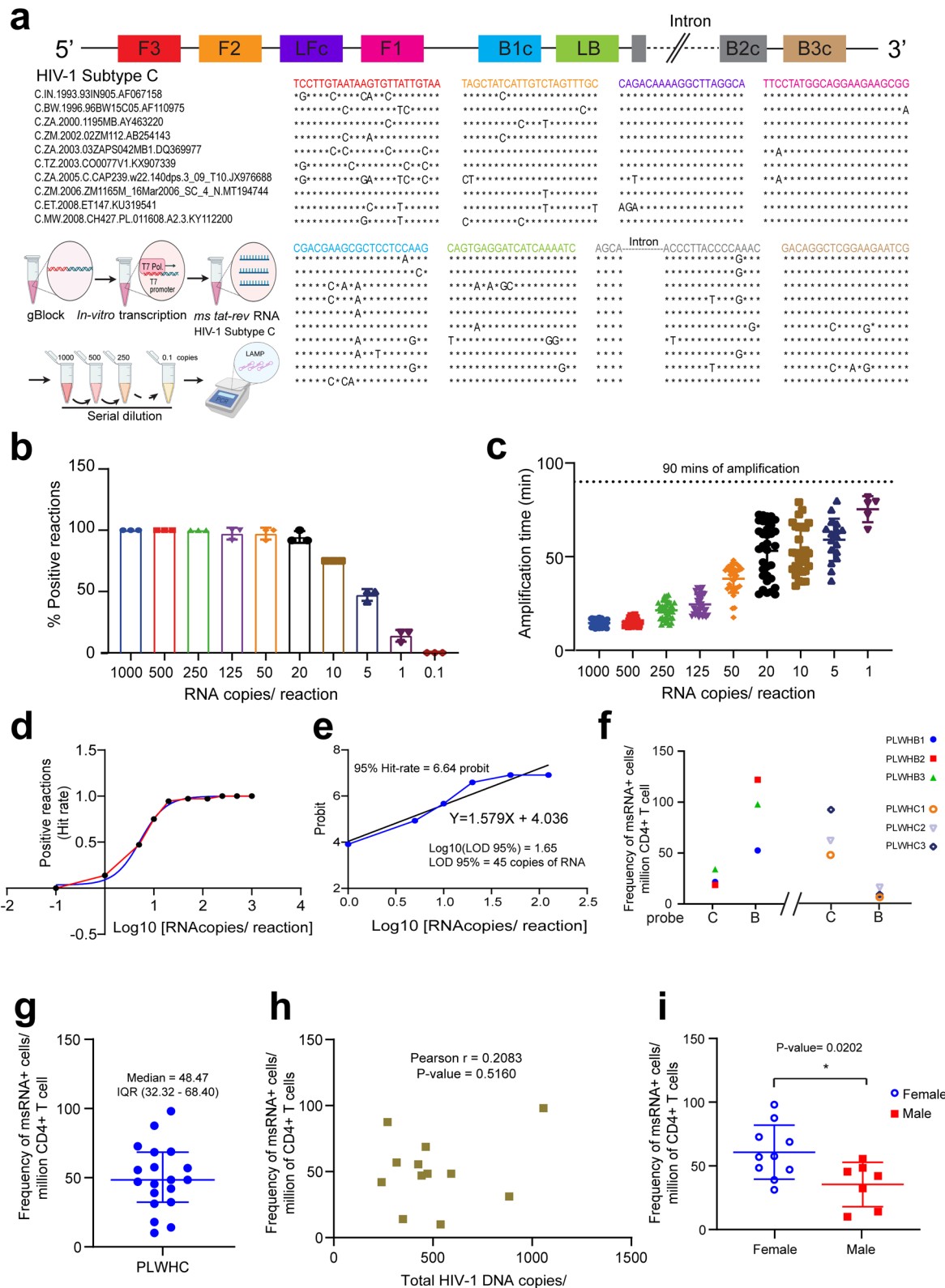

reversal suggested msRNA may be a useful biomarker of virion production[43], with the caveat that in ex vivo interventional models, prolonged pharmacological stimulation of cells could induce a non-physiological state, cell death, or amplification of RNA signals unrelated to replication competence. Nevertheless, the detection of *tat/rev* msRNA transcripts diminishes the probability of measuring defective proviruses

with large internal deletions, as the msRNA transcripts are produced through the splicing of full-length viral transcripts[13]. *tat/rev* msRNA thus holds potential as a relevant indicator for the inducible viral reservoir.

Current methods for *tat/rev* msRNA quantification involve real-time RT-qPCR or RT-ddPCR on bulk total cellular RNA, both highly sensitive and specific for the targeted viral RNA transcripts[36,39]. Certain RNA

**Fig. 6 | Design and application of SQuHIVLa to quantify the inducible reservoir of People with HIV-1 subtype C. a** Ten HIV-1 subtype C viral sequences are aligned with different subtype C-specific LAMP primers. After alignment, the "*" symbol is used to denote identical nucleotides; otherwise, the actual nucleotide symbol (A, T, C, or G) is used. In vitro-transcribed synthetic subtype C msRNA is serially diluted to achieve samples containing a range of target RNA copies used to perform RT-LAMP with subtype C-specific primers and probes. **b** Percentage of positive RT-LAMP reactions and **c** amplification time required for different amounts of RNA copies are plotted. Data are presented as mean ± SD of three independent in vitro transcription experiments followed by serial dilution and RT-LAMP. Twelve reactions were performed per condition for each independent experiment. Dotted line represents 90 min of isothermal amplification. **d** A non-linear regression analysis is performed using the proportion of positive RT-LAMP reactions (HIT rate) (Y-axis) corresponding to Log10 values of RNA copies used as a template (X-axis). **e** Probit analysis is performed to determine LOD-95% using probit values calculated from the HIT rate (Y-axis) corresponding to Log10 values of RNA copies (Y-axis). **f** The inducible reservoir was quantified for 3 PLWHB and 3 PLWHC using both subtype-specific (e.g. subtype B and subtype C-specific primers/prob for PLWHB and PLWHC respectively) and non-specific LAMP primers/probe (e.g. subtype B and subtype C-specific primers/prob for PLWHC and PLWHB respectively) to the determine the specificity of primers/probe towards their respective HIV-1 subtype. **g** The inducible HIV-1 reservoir was quantified for $N = 19$ people living with HIV-1 subtype C (PLWHC) presented as median with interquartile range. **h** Pearson ($r$) correlation between SQuHIVLa and total HIV-1 DNA copies for 12 PLWHC. Statistical significance determined by $p < 0.05$. **i** The inducible HIV-1 reservoir was quantified for ten female (blue open circle) and seven male (red square) PLWHC using SQuHIVLa assay. Data are presented as mean ± SD. Two-tailed unpaired $t$-test is performed to analyze the variation of inducible HIV-1 reservoir between male and female PLWHC and statistical significance is determined by $p < 0.05$.

induction assays directly probe *tat/rev* msRNA in whole cells without RNA extraction, simplifying the workflow to perform semi-nested RT-qPCR[35,46]. However, employing the direct whole cell input without DNase digestion, followed by semi-nested RT-qPCR, as done in previously developed assays like TILDA[35,65,73–75], presents limitations that may impact assay accuracy. Notably, this method increases the risk of detecting *tat/rev* within genomic HIV DNA due to the lack of usage of primers and probes that span exon junctions to exclusively amplify *tat/rev* msRNA, potentially causing false positives and overestimating measurements of the inducible viral reservoir. Moreover, semi-nested RT-qPCR approaches such as TILDA, performed in limiting dilution, have low throughput and considerably increase cross-contamination risks while transferring the pre-amplified PCR products in a new plate for final amplification, the latter leading to false measurements[74]. We developed SQuHIVLA, a specific and sensitive RT-LAMP assay, to overcome the limitations of *tat/rev* msRNA RT-qPCR. This rapid and highly accurate method offers a novel application of LAMP technology for quantifying *tat/rev* msRNA+ HIV-1 viral reservoir.

The swift and highly sensitive detection of *tat/rev* msRNA, surpassing existing semi-nested RT-qPCR methods, represents a groundbreaking advancement. Leveraging the rapid amplification, high sensitivity, and specificity of RT-LAMP in a single reaction, we improved the detection and quantification of cells expressing *tat/rev* msRNA upon activation (SQuHIVLa). As a proof of concept, we demonstrated an RT-LAMP-based assessment of the inducible viral reservoir in CD4+ T cells from individuals with HIV-1 subtypes B and C under suppressive ART. The quantitative nature of the SQuHIVLa, i.e. the number of cells expressing HIV-1 msRNA per million CD4+ T cells, is accomplished through (1) the specific and sensitive detection of HIV-1 msRNA by RT-LAMP followed by (2) maximum likelihood calculations based on the distribution of cells in limiting dilution format. Despite the limited data, our results show promising sensitivity, specificity, and low inter-assay variability. The test exhibited high sensitivity, quantifying the inducible reservoir with as few as 10 cells per million CD4+ T cells, underscoring its effectiveness. Further evaluation with cells from individuals with low or ultra-low viral reservoir sizes is recommended.

In designing LAMP primers exclusively for *tat/rev* msRNA detection, it was crucial to ensure they were exon-spanning. The *tat/rev* intron, approximately 2 kb, if included in the loops during amplification, would destabilize the loops, hindering further target amplification as depicted in Fig. 2a. This instability would render the fluorescent self-quenching loop primer[63], a reporter for specific target amplification, non-functional. We have developed a comprehensive set of guidelines for designing *tat/rev* exon-spanning LAMP primers, meeting target-specific criteria along with standard LAMP primer requirements (Supplementary Note 1). Designing LAMP PCR primers and a probe to bind within the *tat/rev* genomic region is extremely challenging due to high HIV-1 genetic diversity and the need to identify eight relatively conserved primer binding regions for the LAMP reaction. Additionally, our data demonstrated less efficient amplification of *tat/rev* msRNA when subtype B-specific primers/probe were employed for

PLWHC and vice versa, leading to suboptimal quantification of the inducible reservoir (Fig. 6f). Because of this, a universal LAMP primer/probe set for the broad detection of *tat/rev* msRNA expressed from multiple HIV-1 subtypes is quite likely not feasible. Multiplexing LAMP primer/probe sets for different HIV-1 subtypes into one reaction is potentially challenging to achieve[76] without complex molecular design[77,78] and extensive optimizations.

SQuHIVLa demonstrates a significant positive correlation with intact HIV-1 DNA copies but not with total HIV-1 DNA copies. This observation may be attributed to a substantial proportion of the total HIV-1 DNA copies comprising defective proviruses, which may not contribute to the inducible reservoir expressing HIV-1 msRNA. SQuHIVLa exhibited a moderate Spearman's rank correlation coefficient ($r$) with the reservoir size measured by TILDA, although this correlation was not statistically significant. The absence of a robust correlation and statistical significance may, in part, be attributed to the limited sample size. Furthermore, the disparity in amplification methods, along with differences in sensitivity and specificity for the target, as well as the potential for false positive signals from proviral DNA contamination in TILDA, could contribute to these findings.

Interestingly, in our cohort of PLWHC during chronic infection, women exhibited a significantly higher mean frequency of cells expressing *tat/rev* msRNA after PMA/Ionomycin treatment compared to men. Despite limited research, existing studies on HIV-1 reservoir size, primarily assessed through DNA quantification, report comparable levels in men and women[79–81]. Our findings contradict studies focusing on viral transcription and replication (viral outgrowth) that reported a smaller inducible viral reservoir in women[82]. This disparity may rise from differences in endpoints, such as us- versus msRNA transcripts, reflecting varied capacities for viral inducibility. Furthermore, viral outgrowth assay (QVOA) often underestimates reservoirs due to suboptimal provirus induction[83]. It is likely that the frequency of inducible virus is truly small in women, or poorly inducible. Potential sex-specific differences in inducible reservoir size may be influenced by estrogen, known to repress HIV-1 transcription in latency models[84], in vitro infection systems[85], and primary cells[84]. Further research with a larger sample size and mechanistic studies are crucial for comprehending sex-specific implications in reservoir maintenance and dynamics.

Amid the rise of explorative strategies to clear the inducible viral reservoir in PLWH, reliable, sensitive, and scalable assays are essential to determine the efficacy of putative intervention strategies. Moreover, future studies should focus on evaluating large cohorts in Sub-Saharan Africa[86], which bears a disproportionate burden of HIV-1[2]. This emphasizes the necessity for viral reservoir quantification assays applicable in resource-constrained settings[87]. RT-LAMP is expected to contribute to this goal as it is highly sensitive, capable of detecting low copy numbers of RNA targets, and can provide quantitative results through real-time monitoring. Furthermore, it is less expensive, more user-friendly, and compatible with a variety of sample types. These features make RT-LAMP a promising tool for quantitative analysis in various fields, including HIV-1 molecular diagnostics and viral load monitoring, especially in limited resource settings.

However, despite all the advantages, the RT-LAMP approach combined with maximum likelihood calculations also presents a few limitations in quantifying the frequency of msRNA-expressing cells: Firstly, the complexity of LAMP is heightened, particularly in primer design, when aiming to position the exon/exon junction inside the loop for specific msRNA amplification, especially for sequences such as *tat/rev* msRNA, which has a high polymorphic content. Likewise, the maximum likelihood calculation for determining the frequency of inducible HIV-1 reservoirs faces challenges in precisely estimating the true reservoir size due to potential model assumptions and uncertainties in defining latent reservoir parameters. Opting for a single-cell approach like flow cytometry to identify msRNA-positive cells, instead of relying on maximum likelihood statistics, offers a potential solution to challenges associated with the limiting dilution step. Nonetheless, achieving this requires substantial optimization efforts. Therefore, advancing research and fostering collaborations, particularly with biomedical industries, is essential to develop certified high-throughput assays for HIV-1 and other pathogens based on the principles of SQuHIVLa.

## Data availability

All data needed to evaluate the conclusions in the paper are present in the paper and/or the Supplementary Information. All source data used in the manuscript is provided in the Supplementary Data 1 file. Additional data related to this paper may be requested from the authors.

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

## Acknowledgements

T.M. received funding from Health Holland [grants LSHM19100-SGF and EMCLSH19023]; ZonMW [grant 40-44600-98-333] and a Building Synergistic Infrastructure for NL4Cure grant from Dutch Aidsfonds. C.L. received funding from the Dutch Aidsfonds (grants P-60602 and P-263). S.R. received funding from Dutch Aidsfonds [P-53302]. P.M. received funding from the South African Medical Research Council. The funders had no role in the study's design and execution, data collection, management, analysis, interpretation, or the preparation, review, or approval of the manuscript.

## Author contributions

Conceptualization, T.H., C.L., and T.M.; Methodology, T.H., C.L., and T.M.; Investigation, T.H., C.L., S.D.S., M.K., R.C., N.R., A.N., T.W.K., and K.R.; Funding Acquisition, T.M., S.R., C.L, P.M., and T.N.; Project Administration, T.M., S.R., R.J.P., P.M., K.R., and T.N.; Supervision, T.M.; Writing – Original Draft, T.H., C.L., S.D.S, M.K., and T.M.; Writing – Review & Editing, T.H., C.L., S.D.S, M.K., R.C., N.R, A.N., T.W.K., K.R., S.R., R.J.P., P.M., T.N., and T.M.

## Competing interests

The authors declare the following competing interests: T.H., C.L., and T.M. are listed as inventors on the patent application filed by the Erasmus University Medical Center (EP23183103) on the methods for specific quantitation of the inducible HIV reservoir. The other authors declare no competing interests.
