## [Peer Review File · Communications Medicine]

Reviewers' comments:

Reviewer #1 (Remarks to the Author):

The manuscript describes an induction assay, as far as I understand is derived from TILDA, with a different detection method as read-out after activation of HIV+ cells. The work-flow described here very closely follows the Procopio et al. 2015 paper but uses LAMP instead of RT-qPCR for the detection of multiply spliced RNA as surrogate for the inducible HIV reservoir. However, the manuscript could benefit from careful revision and some additional experiments.

General comments:

1. In the heading the word “novel” should be replaced with either “alternative” or “faster”.
2. Throughout the authors use “inducible” and “replication-competent” HIV almost interchangeably when talking about induction of msRNA. But inducible virus is not necessarily replication competent, even though correlations have been found by different publications. The terminology should be adjusted where necessary. Furthermore, the assay presented in this manuscript does not detect replication-competent virus, only the inducible reservoir.
3. The authors mention in several places that their msRNA-LAMP assay could be easier rolled out in LMICs due to easier workflow and faster read-out compared to other assays. However, it is not clear in what way this could be achieved since cell selection and cell culture is still necessary and takes time (<12 hours) with only the detection method being faster. The rationale for this is missing and should be included.
4. The Introduction is quite long and the Methods are very repetitive. Both could be shortened significantly.
5. The authors talk about exon- and non-exon-spanning primers. It is unclear what is meant by this, the TILDA-primers used for qRT-PCR, as well as the LAMP primers described here are intron-spanning. This should either be clarified or corrected in the text.
6. The quantitative nature of the assay is not clear. Does this refer to RNA copy numbers that can be detected per well or the msRNA positive wells (inducible reservoir) in the limiting dilution? This should be clarified.

Abstract:

7. The authors state that their assay “exclusively detects subtype-specific HIV-1 ...” but they only show results for subtypes B and C. This should be clarified. Additionally, it could help to switch this one and the following sentence.

Introduction:

Minor comments:

8. Line 49: Rebound occurs in a range of several weeks, with a mean of two weeks. Changing the wording would make this statement more accurate.
9. Line 53/54: Do the authors talk about the total, inducible or latent viral reservoir?
10. Lines 64-66: Sentence should be re-phrased because it is very difficult to understand.
11. Line 129/130: "Binding" should be used instead of "targeting". "Targeting" would imply a different gene or genomic region of HIV where the primers can bind.
12. Lines 133-136: The described assay in this manuscript is not entirely new, since only the detection method differs to previously used assays (TILDA). This statement should be adjusted.

Methods:

Major comments:

13. The majority of the methods (especially cell isolation, cell activation, use of limiting dilution) have been taken from Procopio et al. 2015 in very similar wording but without adequate citation. This needs to be corrected and the reference needs to be stated clearly.
14. The methods are very repetitive (cell activation, LAMP conditions,...) and can be shortened considerably.
15. In Table 1 the subtypes for the different PLWH should be added. Also, if the total HIV DNA reservoir of these patients is known this could be added in the table and compared in the Results to the inducible reservoir.
16. line 211 and 222: How were the GFP+ cells sorted into the CD4 background cells? Which volume was put in the PCR reaction? Were the cells in RPMI or a buffer?
17. Line 257: How was the inducible reservoir of these samples determined before testing them in the LAMP assay?

Minor comments:

18. Line 157: Please specify the subtype
19. Line 188: How were the copy numbers of the control RNA determined?
20. Starting at line 221: Comparison of LAMP and qPCR should go in Results chapter.

Results:

Major comments:

21. For description of Figure 1 it would be beneficial to add numbers or letters in the Figure which can be referred to in the text, for easier comprehension. Also, since reading direction is usually from top left to bottom right flipping the direction of the figure would make it easier to understand.
22. Lines 339/340: Due to the high sensitivity of LAMP did the authors find a Ct/time (in minutes) and RNA concentration where a sample is still considered negative? Even negative reactions can become positive after sufficient incubation time.
23. Were the LAMP primers for subtype B and C tested for cross-reactivity for the two subtypes (or were other subtypes tested)? This should be done for determination of specificity or perhaps broader ability of HIV detection.
24. Lines 391-393: Why was the incubation step of 45 degrees added? What is the effect/rational of this?
25. The authors describe and use the primers for the qRT-PCR from the Procopio et al. 2015 paper. How does the inducible reservoir with this method compare to their LAMP assay? It would be good to do these experiments on the HIV patient samples from subtypes B and C to see whether sensitivity and specificity of the LAMP assay is comparable.
26. When different cell numbers for samples are described it is not clear whether the different cell concentrations were further diluted in a limiting dilution or whether they are already defined as the limiting dilutions.

Minor comments:

27. Figure 2: The 5'-3' direction of the primers should be indicated with arrows.
28. Line 334: The fluorophore should be added in the text. Related to this, in Suppl. Table 2 the authors only state the fluorophore. I am not familiar with i6-FAMK-probes, does it not need a quencher?
29. Line 365: Are cells not expressing msRNA defined as not activated cells?
30. Lines 381-383: The sentence should be re-phrased for better understanding.
31. Line 422: Was the duration of infection before ART initiation known? If so, is this correlated with the size of the inducible reservoir?

Discussion:

Major comments:

32. As with Introduction and Methods this chapter is quite extensive and could be shortened. Some parts (like the beginning) would be better placed in the Introduction.

33. Overall, the statements the authors make are quite strong and, I feel, not entirely justified by their results. While the use of intron-spanning LAMP primers is a very interesting approach, the main difference between their assay and the already used TILDA is only a faster read-out after cell activation, which saves approximately 1.5 to 2 hours of time compared to qRT-PCR. Does this justify the HIV-LAMP assay being superior? Comparison between RT-qPCR and LAMP is not shown. Furthermore, how are “high specificity and sensitivity” defined for the LAMP assay? This was not entirely clear from the results.

Reviewer #2 (Remarks to the Author):

The authors have developed a novel approach for quantitating the inducible HIV reservoir using reverse transcription loop-mediated isothermal amplification (RT-LAMP) of multiply spliced HIV RNA. The paper is well written, with an excellent introduction and extensive validation, but could be improved by considering the following points

1) The major problem with the paper is that the authors do not provide any comparative data to let the reader judge how the assay compared to previously published reservoir assays.

2) While the assay may have certain practical advantages over other assays, it is in principle similar to the TILDA assay. It would be interesting to see the assays compare.

3) The new assay gives infected cell values that are much higher than those observed with the gold standard QVOA assay but lower than the IPDA. Can the authors explain this?

4) It would be helpful if the authors could explain in what situations the new assay would be particularly useful.

Reviewer #3 (Remarks to the Author):

The Mahmoudi group presents a crystal-clear and novel protocol for assessing the number of HIV RNA transcripts expressed in a pool of CD4+ cells, in the presence or absence of exogenous stimulation. They

argue that this protocol is simpler, cheaper, faster, and less prone to HIV DNA contamination than other such protocols. It is promising that clades other than Clade B can be assessed with this technique.

As presented, this report may be useful for the field, but its true impact will depend on how widely it is tested and compared to other "HIV Reservoir" assessment tools. The work could be improved by several additions:

- "inducible reservoir size" (eg. Fig 4): the word reservoir is widely misused. What is shown is "frequency of RNA+ cells. All measures of the reservoir (persistent proviruses that can result in infectious rebound viremia off ART) are imperfect and/or surrogate measures. The data in Fig 4 is presented in isolation, and any comparison (with concurrent measures of HIV DNA, of IPDA or q4PCR, of QVOA or dQVOA, or most especially of TILDA, would improve the value of this work. The authors work at an HIV center, and there should be samples available in someone's freezer that have already been assessed by other measures.
- an incomplete discussion of risk of DNA detection in semi-nested RT-qPCR approaches (eg TILDA): this reservoir is placed in the same niche as TILDA, and a frank discussion of the problems of TILDA would improve this work
- an incomplete discussion of the use of ms-RNA as a surrogate measure: the Zerbato paper is deeply flawed, as cells are stimulated for 3 days --- a non-physiological state which results in the loss of some cells (death) and the amplification of other RNA signals that may or may not be RC viruses. ms-RNA may be useful surrogate, but it is only a surrogate, and the correlations present in the literature (like those for TILDA) are suboptimal. There is no final answer for these controversies, but a better discussion would be helpful.
- the assay used CD4+ cells that are negatively selected. Can it be performed on unselected PBMCs? As a measure of cells with inducible HIV RNA, PBMC assays might be little different and much cheaper to run.

Reviewer #1 (Remarks to the Author):

The manuscript describes an induction assay, as far as I understand is derived from TILDA, with a different detection method as read-out after activation of HIV+ cells. The work-flow described here very closely follows the Procopio et al. 2015 paper but uses LAMP instead of RT-qPCR for the detection of multiply spliced RNA as surrogate for the inducible HIV reservoir. However, the manuscript could benefit from careful revision and some additional experiments.

General comments:

1. In the heading the word “novel” should be replaced with either “alternative” or “faster”.

We have now removed the word “Novel” from the title of the manuscript which is modified to: “SQuHIVLa: Specific Quantification of inducible HIV-1 reservoir by LAMP”, according to the reviewer’s suggestion.

2. Throughout the authors use “inducible” and “replication-competent” HIV almost interchangeably when talking about induction of msRNA. But inducible virus is not necessarily replication competent, even though correlations have been found by different publications. The terminology should be adjusted where necessary. Furthermore, the assay presented in this manuscript does not detect replication-competent virus, only the inducible reservoir.

We thank the reviewer for pointing this out. In the revised manuscript, we have changed the terminology to “inducible” instead of “inducible replication-competent”, for example see lines no. 29, 56, 483, 551.

3. The authors mention in several places that their msRNA-LAMP assay could be easier rolled out in LMICs due to easier workflow and faster read-out compared to other assays. However, it is not clear in what way this could be achieved since cell selection and cell culture is still necessary and takes time (<12 hours) with only the detection method being faster. The rationale for this is missing and should be included.

We acknowledge the reviewer's observation that, like many assays, SQuHIVLa requires cell isolation and culturing. However, we contend that SQuHIVLa offers advantages for potential implementation in low- to middle-income countries (LMICs) because of the following reasons which as suggested by the reviewer we now also clarify in the manuscript (lines 107-112 ; 562-567). First, the use of RT-LAMP in SQuHIVLa as a single-tube reaction reduces instrument constraints, distinguishing it from semi-nested RT-qPCR. Second, this single-tube reaction feature also enhances SQuHIVLa's resilience against cross-contamination which can result in false positive signals. Third, the reagents used in RT-LAMP, in particular the enzymes used, are comparatively more cost-effective than those used in standard semi-nested RT-qPCR assays. In addition, we extensively point out in the discussion section that our RT-LAMP HIV-1 msRNA method is potentially amenable to single-cell approaches such as flow cytometry for quantification, instead of relying on maximum likelihood statistics, thereby further reducing costs associated with the limiting dilution step while increasing accuracy (discussed in line no. 575-578). Considering these factors, we

believe that SQuHIVLa demonstrates significant scalability potential compared to other reservoir quantitation assays and is better-suited for implementation in resource-constrained, LMIC settings.

4. The Introduction is quite long and the Methods are very repetitive. Both could be shortened significantly.

As suggested, we have made substantial revisions to the introduction and methods sections. The revised introduction is now 277 words shorter than the original manuscript, reflecting a more concise presentation. The method section has undergone extensive modifications, incorporating detailed clarifications and additional information based on the reviewer's suggestions. Notably, new content has been included on performing IPDA, TILDA, and total HIV-1 DNA quantification by ddPCR. Despite these additions, the revised manuscript is only 173 words longer than the previous version.

5. The authors talk about exon- and non-exon-spanning primers. It is unclear what is meant by this, the TILDA-primers used for qRT-PCR, as well as the LAMP primers described here are intron-spanning. This should either be clarified or corrected in the text.

Exon-spanning implies the presence of primers with one end mapping to one exon, and the other end mapping to another exon, thus spanning the intermediate intron. Intron-spanning primers sit within different exons while spanning a large intron. We clarify this in the supplementary notes section. The TILDA assay, developed by Procopio et al.¹, employs an intron-spanning set of primers and a probe for both pre-amplification and final amplification in their semi-nested RTqPCR assay (Supplementary Figure 2C). An inherent limitation when utilizing intron-spanning primers in samples containing both proviral HIV-1 DNA and msRNA, without a dedicated step for DNA degradation (e.g., DNase treatment), as observed in both TILDA and SQuHIVLa, is the potential for false-positive results originating from proviral HIV-1 DNA rather than msRNA.

In the Procopio et al. (2015) study, the thermal protocol for pre-amplification includes a 4-minute step at 60°C (for annealing and elongation), suitable for amplifying the target sequence with an approximately 2-kb long intron in proviral HIV-1 DNA. This condition may potentially contribute to a positive signal in the absence of msRNA (as we also show in supplementary figure 2B). Therefore, the use of exon-spanning primers, ensuring exclusive amplification of only msRNA, is more suitable in this specific scenario. Exon-spanning primers result in an unbound 3' end of the primer, preventing amplification initiation in the presence of an intron (this is demonstrated schematically in Figure 2A).

LAMP primers function differently from those used in conventional PCR. Unlike conventional PCR, where amplification initiates independently from both forward and reverse primers, forward outer and inner primers in LAMP work collaboratively through sequential binding and initial complementary strand elongation to form a forward loop, and vice versa for a backward loop with backward outer and inner primers. Successful LAMP amplification requires the formation of a stable dumbbell structure with forward and backward loops. In this context, forward and backward loops (not the individual primers) serve a function similar to forward and reverse primers in conventional amplification. Strategic placement of the exon/exon boundary in either the forward or backward loop ensures that the presence of an intron leads to an unstable forward or backward loop, as depicted in Figure 2, thereby preventing amplification of proviral HIV-1 DNA. Consequently, the LAMP primers used in this manuscript are referred to as exon-

spanning.

6. The quantitative nature of the assay is not clear. Does this refer to RNA copy numbers that can be detected per well or the msRNA positive wells (inducible reservoir) in the limiting dilution? This should be clarified.

This is an important point, which we now better clarify in the text (lines 382-409). The quantitative nature of the assay SQuHIVLa refers to the number of cells expressing HIV-1 msRNA per million CD4+ T cells. This is accomplished through 1) the specific and sensitive detection of HIV-1 msRNA by RT-LAMP followed by 2) maximum likelihood calculations based on the distribution of cells in limiting dilution format. However, as the utilization of RT-LAMP for detecting HIV-1 msRNA is a new approach, it was critical to first perform a systematic validation to assess the specificity and analytical sensitivity of RT-LAMP in detection of Tat/Rev msRNA (depicted in Fig. 3 and Fig. 4A-E). Observing satisfactory analytical sensitivity, we transitioned the goal of quantifying the number of inducible msRNA-positive cells per million CD4+ T cells. Here we determined the number of cells and reaction conditions for use in the limiting dilution format and conducted SQuHIVLa to quantify the frequency of msRNA-expressing cells during induction (inducible reservoir) across participants infected with HIV-1 subtypes B (depicted in Fig. 4F-H) and C (depicted in Fig. 6G). We hope this clarification provides a better understanding of the quantitative nature of SQuHIVLa.

Abstract:

7. The authors state that their assay “exclusively detects subtype-specific HIV-1 ...” but they only show results for subtypes B and C. This should be clarified. Additionally, it could help to switch this one and the following sentence.

We appreciate the reviewer's suggestion. While we have presented data specific to HIV-1 subtypes B and C in this manuscript, we have also included a comprehensive step-by-step protocol (depicted in Fig. 1, detailed in the Methods section, and provided in Supplementary Notes) for designing LAMP primers and probes tailored to any HIV-1 subtype. The subtype B and C primers and probes presented here were developed by following this protocol sequentially. In light of this clarification, we have revised the wording in lines 32-34 of the revised manuscript.

Introduction:

Minor comments:

8. Line 49: Rebound occurs in a range of several weeks, with a mean of two weeks. Changing the wording would make this statement more accurate.

We agree. This has been rephrased in lines 47-50 of the revised manuscript.

9. Line 53/54: Do the authors talk about the total, inducible or latent viral reservoir?

To be inclusive of intervention approaches that do not necessarily target the inducible reservoir we have changed this to 'latent viral reservoir' (line 53), thank you for pointing this out.

10. Lines 64-66: Sentence should be re-phrased because it is very difficult to understand.

As per the reviewer's suggestion, we have rephrased the sentence to "By targeting a single highly conserved proviral region, defective HIV-1 genomes with substantial deletions or hypermutations in places other than the test amplicon can also be detected" in lines 64-66 of the revised manuscript.

11. Line 129/130: "Binding" should be used instead of "targeting." "Targeting" would imply a different gene or genomic region of HIV where the primers can bind.

We have incorporated the suggested change from 'targeting' to 'binding' in line 113 of the revised manuscript.

12. Lines 133-136: The described assay in this manuscript is not entirely new, since only the detection method differs to previously used assays (TILDA). This statement should be adjusted.

We have changed this sentence to "Here, we present SQuHIVLa (Specific Quantification of Inducible HIV-1 reservoir by LAMP), a RT-LAMP-based assay to detect cells expressing Tat/Rev mRNA, addressing the gap in HIV-1 reservoir quantification" (lines 117-119).

Methods:

Major comments:

13. The majority of the methods (especially cell isolation, cell activation, use of limiting dilution) have been taken from Procopio et al. 2015 in very similar wording but without adequate citation. This needs to be corrected and the reference needs to be stated clearly.

We acknowledge the similarity of parts of the methods section pertaining to limiting dilution and maximum likelihood calculation to the previously published study by Procopio et al. 2015. This is mainly due to the fact that SQuHIVLa uses a similar approach for isolation and stimulation of the primary CD4+ T cells as well as limiting dilution for the quantification of the inducible reservoir. We have made extensive modifications to the wording and included appropriate citations to the original TILDA study by Procopio et al. where applicable.

14. The methods are very repetitive (cell activation, LAMP conditions,...) and can be shortened considerably.

As recommended by the reviewer, the method section has been extensively modified to ensure more concise presentation while providing more clarifications and addition of new information.

15. In Table 1 the subtypes for the different PLWH should be added. Also, if the total HIV DNA reservoir of these patients is known this could be added in the table.

We have updated Table 1, which now includes the size of total HIV-1 DNA reservoir measured by ddPCR.

16. line 211 and 222: How were the GFP+ cells sorted into the CD4 background cells? Which volume was put in the PCR reaction? Were the cells in RPMI or a buffer?

We now explain in detail the sorting procedure in lines 190 to 205 of the revised manuscript. Briefly, the GFP+ cells and uninfected donor-derived primary CD4+ T cells were sorted by adding a single GFP+ J-Lat 11.1 cell directly into each well of a 96-well PCR plate. The PCR plate contained 20 μ L of the RT-LAMP master mix, either without background or with an increasing background of activated uninfected donor CD4+ T cells. The final reaction volume was maintained at 20 μ L after the single cell sorting as the sorted volume is negligible. The activated uninfected donor-derived primary CD4+ T cells and J-Lat 11.1 cells were washed in RPMI 1640 media supplemented with 3% FBS before resuspension in PBS for sorting.

17. Line 257: How was the inducible reservoir of these samples determined before testing them in the LAMP assay?

The samples under discussion are custom samples prepared to represent clinical samples with a predefined msRNA+ inducible reservoir size. These custom samples were generated using a known quantity of GFP+ J-Lat 11.1 cells as a surrogate of msRNA+ inducible reservoir and healthy donor CD4+ T cells as a surrogate for an uninfected cell background. The detailed procedure for preparing these custom samples is outlined in lines 229 to 238 of the revised manuscript and illustrated in Fig. 4C.

Minor comments:

18. Line 157: Please specify the subtype

HIV-1 subtype is now specified in line 142 of the revised manuscript.

19. Line 188: How were the copy numbers of the control RNA determined?

We note that this important information was indeed missing from the original version of the manuscript. The procedure for determining RNA copy numbers is now detailed and provided in lines 161 to 173 of the revised manuscript. In summary, a gBlock containing a spliced Tat/Rev sequence (subtype B or C) downstream of a T7 promoter was transcribed in vitro using Hi-T7 RNA Polymerase and Ribonucleotide Solution Mix. After transcription, RNase-free DNase I was used to treat and eliminate gBlock DNA, followed by purification. Quantification of each purified Tat/Rev RNA sample was performed five times using a NanoDrop 200 spectrophotometer. The average values, along with respective lengths (509 bp for subtype B and 483 bp for subtype C Tat/Rev RNA), were used to determine the RNA copies per microliter.

Subsequently, the samples were serially diluted to concentrations ranging from 1000 to 0.1 copies of RNA per 5 μ L for use in sensitivity validation experiments.

20. Starting at line 221: Comparison of LAMP and qPCR should go in Results chapter.

As per the reviewer's suggestion, we have updated the result section and the comparison between RT-LAMP and semi-nested RT-qPCR is now mentioned in lines 366 to 371 of the revised manuscript.

Results:

Major comments:

21. For description of Figure 1 it would be beneficial to add numbers or letters in the Figure which can be referred to in the text, for easier comprehension. Also, since reading direction is usually from top left to bottom right flipping the direction of the figure would make it easier to understand.

Thank you for this suggestion. The revised Fig. 1 now reads from the top left to the bottom right, and each step in the corrected Fig. 1 has been labeled with a letter (e.g., step a, b, c, etc.), enhancing clarity in the flowchart. These labels are also referred to in the text within the results section for improved comprehension.

22. Lines 339/340: Due to the high sensitivity of LAMP did the authors find a Ct/time (in minutes) and RNA concentration where a sample is still considered negative? Even negative reactions can become positive after sufficient incubation time.

Experiments presented in Figure 3C and 6C were performed using 120 minutes of amplification. These results indicated that all positive signals, even for a single copy of RNA, were detected before 90 minutes of amplification. Therefore 90 minutes of amplification was used for further experiments. Based on the reviewer's comment, we have performed SQuHIVLa for a well characterized participant using 120 minutes of isothermal amplification instead of 90 minutes in order to determine any potential change of reservoir size quantified when longer amplification is performed. The reservoir size did not change with longer amplification and the positive signals were achieved before 90 minutes of amplification, consistent with previous observations. We have included this data in suppl. Fig. 4. This observation suggests that the negative samples remained negative at least up until 120 minutes of amplification.

23. Were the LAMP primers for subtype B and C tested for cross-reactivity for the two subtypes (or were other subtypes tested)? This should be done for determination of specificity or perhaps broader ability of HIV detection.

We thank the reviewer for this suggestion. To assess the cross-reactivity of the LAMP primers/probe, inducible reservoirs were quantified in three individuals with HIV-1 subtype B (PLWHB) and three with subtype C (PLWHC). Both subtype-specific (subtype B and subtype C specific primers/probe for PLWHB and PLWHC, respectively) and non-specific (subtype C and subtype B specific primers/probe for PLWHB and PLWHC, respectively) LAMP primers/probe sets were used. We now present this data in the revised manuscript in Fig. 6F and in lines 449 to 456. The results indicated suboptimal reservoir quantification

when subtype non-specific primers/probe were employed, emphasizing the necessity of designing subtype-specific primers/probe sets for optimal reservoir quantitation.

24. Lines 391-393: Why was the incubation step of 45 degrees added? What is the effect/rational of this?

The efficient release of RNA is critical for successful amplification as whole cells are used as input for amplification, and we achieved this by incorporating a high concentration of Triton-X100 into the RT-LAMP reaction mix. While this approach worked well for J-Lat 11.1 cells, we encountered challenges with primary CD4+ T cells, likely attributed to their smaller size and lower RNA content compared to immortalized cell lines. Recognizing the need for improved RNA release, we focused on optimizing this process. Given that the efficiency of Triton X-100 in membrane permeabilization is influenced by factors such as concentration, incubation time, and temperature, we deliberately included a relatively high Triton-X100 concentration in our reaction mix. To enhance sensitivity further, we explored the use of high temperature during incubation. We selected 45 degrees as the temperature, taking into account the WarmStart® enzymes employed for amplification, which ensures the release of the aptamer above this threshold without triggering significant enzymatic activity during the incubation period. The optimization process involved determining the appropriate duration for incubation, and we found that 60 minutes yielded optimal results, as illustrated in Fig. 4G and described in lines 400-407 in the revised manuscript. This optimized protocol was subsequently integrated into the SQuHIVLa methodology.

25. The authors describe and use the primers for the qRT-PCR from the Procopio et al. 2015 paper. How does the inducible reservoir with this method compare to their LAMP assay? It would be good to do these experiments on the HIV patient samples from subtypes B and C to see whether sensitivity and specificity of the LAMP assay is comparable.

We now present a comparative analysis between SQuHIVLa and TILDA in the revised manuscript. The comparison involved assessing SQuHIVLa values alongside TILDA values generated using protocols by Procopio et al.¹ and Lungu et al.², which are recognized for their good reproducibility and low inter-laboratory variability. The analysis encompassed SQuHIVLa values compared with TILDA_Procopio values for 4 PLWHB and TILDA_Lungu values for 9 PLWHB (Fig. 5A-E) (lines 411-426). The results revealed a moderate correlation between SQuHIVLa and TILDA, although statistical significance was not achieved, potentially due to the limited sample size. Disparities in amplification methods, sensitivity, specificity, and the potential for false positives from proviral DNA contamination in TILDA could contribute to these findings (lines 536-541). While we successfully conducted this comparison for subtype B samples, unfortunately, due to the unavailability of samples, we were unable to extend the analysis to subtype C. We appreciate the reviewer's understanding of this limitation.

26. When different cell numbers for samples are described it is not clear whether the different cell concentrations were further diluted in a limiting dilution or whether they are already defined as the limiting dilutions.

We appreciate the opportunity to clarify this point. In Figure 3H-I (line 365), the cell number refers to the number of cells included to determine the maximum background acceptable for specific and sensitive detection of HIV-1 msRNA. In Figure 4C-E (lines 386-393), the cell numbers refer to the number of GFP

positive cells per million CD4+T cells as the custom test samples. Each custom test sample is then used in limiting dilution to quantify the defined reservoir (Figures 4 D-E).

Minor comments:

27. Figure 2: The 5'-3' direction of the primers should be indicated with arrows.

We have indicated the 5' to 3' direction of the primers with arrows in the revised Fig. 2.

28. Line 334: The fluorophore should be added in the text. Related to this, in Suppl. Table 2 the authors only state the fluorophore. I am not familiar with i6-FAMK-probes, does it not need a quencher?

The probes utilized in this study deviate from conventional probes by employing a self-quenching design, eliminating the need for a dedicated quencher. The self-quenching probe, I6-FAMK, specified in Supplementary Table 2, involves a specific code for an internal FAM fluorophore used by IDT, where FAM is linked to the oligo through a dT base using click chemistry. The unique design of self-quenching probes ensures signal quenching of the FAM fluorophore in an unbound state. This approach is chosen due to the challenges posed by the isothermal LAMP amplification procedure and the complexity of primer design, rendering conventional fluorophore-quencher combinations unsuitable. Specific requirements that a self-quenching probe must fulfill are mentioned in detail in the supplementary notes section. Briefly, we have to ensure that the probe sequence has a cytosine (C) or guanine (G) residue at the terminal 3' end, a thymine (T) residue at the second or third position from this 3' end, and optionally, one or more flanking G nucleotides as also mentioned in a previous study by Gadkar et al. 2018³. To clarify, the fluorophore (FAM) has been added to the text in line 336 of the revised manuscript and discussed in supplementary notes.

29. Line 365: Are cells not expressing msRNA defined as not activated cells

We used J-Lat 11.1 cells for validation as a representative model of latent HIV-1 infection. These cells harbor a full-length HIV-1 sequence with the Nef viral gene replaced by a GFP reporter gene (illustrated in Supplementary Fig. 1C). Upon activation with PMA, J-Lat 11.1 cells express viral RNA and protein, with GFP serving as a marker for active viral gene expression, particularly msRNA production as the GFP itself is also a product of the msRNA. However, a distinct subset of PMA-stimulated J-Lat 11.1 cells fail to express GFP, indicating the absence of msRNA production (depicted in Supplementary Fig. 1D).

In the discussed experiment, we specifically utilized this subset of PMA-stimulated J-Lat 11.1 cells lacking GFP expression (referred to as GFP- J-Lat 11.1 cells in Line 373 of the revised manuscript). These cells serve as a surrogate for those containing integrated HIV-1 proviral DNA but lacking msRNA. The objective was to evaluate the discriminatory capability of our designed LAMP primers in distinguishing between msRNA and proviral HIV-1 DNA. We clarify this in the revised manuscript.

30. Lines 381-383: The sentence should be re-phrased for better understanding.

We thank the reviewer for this suggestion. In the revised manuscript we have changed the sentence to “Furthermore, for reservoir sizes greater than 1 GFP+ cell/million CD4+ T cells, the observed and predicted viral reservoir sizes corresponded significantly with good accuracy, with a mean accuracy percentage (AP) of 85%” in lines 391-393.

31. Line 422: Was the duration of infection before ART initiation known? If so, is this correlated with the size of the inducible reservoir?

We now include the probable duration of infection before ART initiation into the patient characteristics table for comprehensive reporting. Based on reviewers’ suggestion, we performed a correlation analysis between the duration of infection before ART initiation and SQuHIVLa values, as shown in Supplementary Fig. 6E. The analysis revealed a weak correlation with a Pearson coefficient (r^2) of 0.07 and a non-significant p-value of 0.3455, which may be attributed to the limited sample size.

Discussion:

Major comments:

32. As with Introduction and Methods this chapter is quite extensive and could be shortened. Some parts (like the beginning) would be better placed in the Introduction.

We appreciate the reviewer's suggestion and have made substantial revisions to the discussion section. The modifications include removing repetitive statements, incorporating new discussion points about correlations of SQuHIVLa with other reservoir quantification assays, and addressing other reviewer suggestions. These changes aim to streamline the discussion and enhance its clarity.

33. Overall, the statements the authors make are quite strong and, I feel, not entirely justified by their results. While the use of intron-spanning LAMP primers is a very interesting approach, the main difference between their assay and the already used TILDA is only a faster read-out after cell activation, which saves approximately 1.5 to 2 hours of time compared to qRT-PCR. Does this justify the HIV-LAMP assay being superior? Comparison between RT-qPCR and LAMP is not shown. Furthermore, how are “high specificity and sensitivity” defined for the LAMP assay? This was not entirely clear from the results.

We acknowledge the reviewer's observation regarding the assay duration. Nevertheless, SQuHIVLa addresses crucial TILDA limitations, such as a higher risk of detecting HIV-1 proviral DNA (supplementary Fig. 2A,B) and cross-contamination during sample transfer for semi-nested RT-qPCR. SQuHIVLa resolves these issues by employing exon-spanning primers exclusively for msRNA and a single-tube RT-LAMP amplification format as discussed above. Moreover, SQuHIVLa uses cost-effective reagents, with potential further reduction in costs when using a single-cell approach like flow cytometry for positive cell identification, instead of the limiting dilution format and maximum likelihood statistics.

With regards to the reviewer’s suggestion to compare RT-qPCR and LAMP, in Supplementary Fig. 2A,B we provide comparison of the two distinct amplification techniques, RT-LAMP and RT-qPCR in their specificity for msRNA detection and the ability to distinguish between msRNA and HIV-1 DNA. Regarding inducible msRNA+ reservoir quantification, we compared SQuHIVLa values with values generated using both TILDA protocols by Procopio et al¹. and Lungu et al.², which have previously been reported to have a very good

reproducibility with low inter-laboratory variability. We compared SQuHIVLa values with TILDA_Procoppio values for 4 PLWHB and TILDA_Lungu values for 9 PLWHB. SQuHIVLa demonstrated a moderate correlation with TILDA, though not statistically significant, possibly due to a limited sample size. Disparities in amplification methods, sensitivity, specificity, and potential false positives from proviral DNA contamination in TILDA may contribute to these findings (Lines 411-426 and 536-541).

Regarding the important point of sensitivity and specificity of RT-LAMP, which we have further clarified in the revised manuscript (lines 343-378), a systematic validation process was conducted to confirm the specificity and sensitivity of RT-LAMP. The primer set's specificity and sensitivity were initially verified using in vitro transcribed msRNA copies, demonstrating a LOD-95% as low as 31 copies (subtype B primers) and 45 copies (subtype C primers) of RNA (Fig. 3A-E). Our designed LAMP primers outperformed RT-qPCR primers in distinguishing between msRNA and HIV-1 DNA. Transitioning to whole-cell input, we used GFP+ J-Lat 11.1 cells as a surrogate for msRNA expression and uninfected donor CD4+ T cells as a negative background. In the absence of CD4+ T cell background, a single GFP+ cell was detected in 95.83% of cases, even at an 83.33% rate with a background of 20,000 cells. Furthermore, we validated the accuracy of RT-LAMP in detecting msRNA-expressing cells using custom experimental samples of predefined inducible reservoir size. These validation experiments informed the determination of cell number and reaction conditions for the limiting dilution format. SQuHIVLa was then conducted to quantify the frequency of msRNA-expressing cells post induction across participants infected with HIV-1 subtypes B (Fig. 4F-H) and C (Fig. 6G).

Reviewer #2

The authors have developed a novel approach for quantitating the inducible HIV reservoir using reverse transcription loop-mediated isothermal amplification (RT-LAMP) of multiply spliced HIV RNA. The paper is well written, with an excellent introduction and extensive validation, but could be improved by considering the following points

1) The major problem with the paper is that the authors do not provide any comparative data to let the reader judge how the assay compared to previously published reservoir assays.

We appreciate the reviewer's feedback and have addressed the concern by providing comparative data in the form of correlation analyses. Specifically, we have conducted correlation analyses of SQuHIVLa values with both total and intact HIV DNA copies, using the well-established Intact Proviral DNA Assay (IPDA) for 12 PLWHB (Fig. 5F,G) (lines 427-432). The results reveal a stronger positive correlation between SQuHIVLa values and intact HIV-1 DNA copies than with total HIV-1 DNA copies. This distinction is significant, as defective proviruses within total HIV DNA copies may not contribute significantly to the inducible reservoir, while intact HIV-1 proviruses are more likely to play a role (discussed in lines 533-536).

Additionally, we performed a correlation analysis of SQuHIVLa values with total HIV DNA copies quantified by ddPCR for 12 PLWHC, showing a similar positive correlation as observed for PLWHB (Fig. 6H) (lines 461-464). These correlation analyses provide valuable insights into how SQuHIVLa compares to established

assays in quantifying different aspects of the HIV reservoir, enhancing the reader's understanding of the assay's performance.

2) While the assay may have certain practical advantages over other assays, it is in principle similar to the TILDA assay. It would be interesting to see the assays compare.

We appreciate the reviewer's suggestion, and to address this comment, we conducted a comparative analysis between SQuHIVLa and TILDA assays. The comparison involved assessing SQuHIVLa values alongside TILDA values generated by Procopio et al.¹ and Lungu et al.², which are recognized for their good reproducibility and low inter-laboratory variability. The analysis encompassed SQuHIVLa values compared with TILDA_Procopio values for 4 PLWHB and TILDA_Lungu values for 9 PLWHB (Fig. 5A-E) (lines 411-426). The results revealed a moderate correlation between SQuHIVLa and TILDA, although statistical significance was not achieved, potentially due to the limited sample size. Disparities in amplification methods, sensitivity, specificity, and the potential for false positives from proviral DNA contamination in TILDA could contribute to these findings (lines 536-541).

3) The new assay gives infected cell values that are much higher than those observed with the gold standard QVOA assay but lower than the IPDA. Can the authors explain this?

We appreciate the reviewer's observation and would like to provide an explanation for the differences observed between SQuHIVLa, QVOA, and IPDA assays. The variation in infected cell values may arise from the distinct methodologies and principles underlying each assay.

SQuHIVLa, while providing a rapid and sensitive means of quantifying inducible HIV-1 reservoirs through RT-LAMP, may inherently detect a broader spectrum of inducible cells compared to QVOA. QVOA, being a gold standard, relies on viral outgrowth from resting CD4+ T cells and may underestimate the reservoir by excluding certain cell populations or viral transcripts that SQuHIVLa can detect.

The differences with IPDA, which quantifies integrated and total HIV DNA, could stem from the fact that SQuHIVLa specifically targets inducible, multiply spliced RNA transcripts, providing a more direct measure of cells potentially capable of active viral replication upon induction. On the other hand, IPDA captures a broader picture of the total HIV DNA pool (intact and defective), which includes integrated and non-integrated forms.

In summary, the variations in infected cell values among the assays are likely due to differences in the cellular and molecular targets they interrogate. Each currently available reservoir quantification assays has its strengths and limitations, and these nuances contribute to the observed distinctions in quantifying HIV-1 reservoirs. We discuss these important differences in the revised manuscript (lines 60-100)

4) It would be helpful if the authors could explain in what situations the new assay would be particularly useful.

Thank you for this suggestion. SQuHIVLa holds promise for several scenarios in HIV research and clinical applications. We have discussed this in lines 485-514 and 558-580 of the revised manuscript. In short, SQuHIVLa offers a faster alternative for assessing inducible HIV-1 reservoirs compared to traditional assays. This advantage is particularly beneficial for high-throughput studies or when timely results are crucial. The assay's design, employing exon-spanning primers for msRNA, enhances specificity, reducing the risk of false positives. The high sensitivity of SQuHIVLa allows for the detection of low-frequency

msRNA-expressing cells, providing a detailed profile of the inducible reservoir. SQuHIVLa employs a single-tube RT-LAMP format and less expensive reagents, contributing to cost-effectiveness. This can be especially advantageous in resource-limited settings or large-scale studies. Due to its rapid turnaround time, SQuHIVLa can facilitate real-time monitoring of interventions targeting the HIV-1 reservoir, allowing researchers and clinicians to assess treatment efficacy more promptly. In summary, SQuHIVLa stands out for its speed, sensitivity, and cost-effectiveness, making it particularly useful in scenarios demanding efficient and accurate assessment of inducible HIV-1 reservoirs, with potential applications in both research and clinical settings. We highlight these points that underline the usefulness of SQuHIVLa as well as its limitations in the current format.

Reviewer #3

The Mahmoudi group presents a crystal-clear and novel protocol for assessing the number of HIV RNA transcripts expressed in a pool of CD4+ cells, in the presence or absence of exogenous stimulation. They argue that this protocol is simpler, cheaper, faster, and less prone to HIV DNA contamination than other such protocols. It is promising that clades other than Clade B can be assessed with this technique.

As presented, this report may be useful for the field, but its true impact will depend on how widely it is tested and compared to other "HIV Reservoir" assessment tools. The work could be improved by several additions:

- "inducible reservoir size" (eg. Fig 4): the word reservoir is widely misused. What is shown is "frequency of RNA+ cells. All measures of the reservoir (persistent proviruses that can result in infectious rebound viremia off ART) are imperfect and/or surrogate measures. The data in Fig 4 is presented in isolation, and any comparison (with concurrent measures of HIV DNA, of IPDA or q4PCR, of QVOA or dQVOA, or most especially of TILDA, would improve the value of this work. The authors work at an HIV center, and there should be samples available in someone's freezer that have already been assessed by other measures.

We appreciate the reviewer's insightful comments and have taken the following actions:

- The term "inducible reservoir size" has been revised to "frequency of msRNA+ cells" or elucidated as cells expressing Tat/Rev msRNA, serving as a surrogate measure for the inducible reservoir, in both figures and texts (where applicable), aligning with the reviewer's suggestion.
- In response to the reviewer's recommendation for comparison with other reservoir quantitation measures, we have conducted correlation analyses. Specifically, we correlated SQuHIVLa values with both total and intact HIV DNA copies using the IPDA assay for 12 PLWHB. The data show a stronger positive correlation with intact HIV-1 DNA copies, emphasizing the relevance of intact proviruses in the inducible reservoir (Fig. 5F,G) (lines 427-432 and 533-536).
- To address the reviewer's inquiry about comparison with TILDA, we performed a comparative analysis. SQuHIVLa values were compared with TILDA values generated using protocols by Procopio et al. and Lungu et al. for 4 and 9 PLWHBs, respectively. While a moderate correlation was observed, statistical significance was not achieved, possibly due to limited sample size and differences in assay methodologies (Fig. 5A-E). These results provide insights into the similarities and distinctions between SQuHIVLa and TILDA (lines 411-426 and 536-541).

We believe these adjustments and additional analyses enhance the clarity and value of our work.

- an incomplete discussion of risk of DNA detection in semi-nested RT-qPCR approaches (eg TILDA): this reservoir is placed in the same niche as TILDA, and a frank discussion of the problems of TILDA would improve this work

Thank you for this suggestion. We address the concern regarding the risk of DNA detection in semi-nested RT-qPCR approaches, specifically in context of TILDA in lines 485 to 497 of the revised manuscript. We make a clear distinction between the use of intron spanning (TILDA) and exon spanning (SQuHIVLa) primer and probe sets. We highlight the limitations associated with semi-nested RT-qPCR methods, including TILDA, in detecting Tat/Rev msRNA. The discussion emphasizes the potential risks of false positives and overestimation of measurements in TILDA due to the detection of Tat/Rev within genomic HIV-DNA.

- an incomplete discussion of the use of ms-RNA as a surrogate measure: the Zerbato paper is deeply flawed, as cells are stimulated for 3 days --- a non-physiological state which results in the loss of some cells (death) and the amplification of other RNA signals that may or may not be RC viruses. ms-RNA may be useful surrogate, but it is only a surrogate, and the correlations present in the literature (like those for TILDA) are suboptimal. There is no final answer for these controversies, but a better discussion would be helpful.

We acknowledge the concerns raised about the potential flaws in the Zerbato paper⁴, pertaining to the use of ms-RNA as a surrogate measure of replication competence, particularly related to the stimulation of cells for three days, which likely induces a non-physiological state leading to cell death and the amplification of other RNA signals that may not indicate replication competence. Nevertheless, previous and emerging work demonstrates potential usefulness of msRNA as a surrogate measure of the reservoir showing correlation with plasma HIV-1 RNA and concurrent expression of msRNA during cellular viral rebound upon ART interruption⁵⁻⁸. We have discussed the arguments for using Tat/Rev msRNA as a surrogate measure while also highlighting the limitations in lines 473-483.

- the assay used CD4+ cells that are negatively selected. Can it be performed on unselected PBMCs? As a measure of cells with inducible HIV RNA, PBMC assays might be little different and much cheaper to run.

We appreciate the reviewer's suggestion to explore the use of unselected PBMCs in the SQuHIVLa assay, which would certainly facilitate application of SQuHIVLa in clinical or resource-constrained settings. To address this, we conducted experiments using total PBMCs instead of negatively selected CD4+ T cells for three participants. The results of the experiment is provided below:

Participant ID	No. of cells/well	No. of wells	No. of positive wells	SQuHIVLa values using PBMC	SQuHIVLa values using CD4+ T cells
PLWHB1	20000	24	1	3.4	54.56
	5000	24	1		
PLWHB2	20000	24	4	8.99	122.81
	5000	24	1		
PLWHB3	20000	24	3	5.27	95.08
	5000	24	0		

The data revealed a substantially smaller reservoir size in total PBMCs compared to CD4+ T cells. While this approach may be feasible, the challenge arises in achieving robust reservoir quantification, especially for low-reservoir samples, due to the limiting dilution and maximum-likelihood calculation inherent in the SQuHIVLa assay. The limited positive signal observed in the PBMC experiments suggests that, to maintain precision in quantifying inducible reservoirs with SQuHIVLa, a significantly larger number of reactions would be required, thereby increasing the cost per sample. This adjustment would compromise the current advantage of SQuHIVLa's suitability for large clinical settings. However, we discuss in the text (lines 575-577), the potential advantage of using single cell approaches such as flow cytometry instead of limiting dilution for the detection of msRNA expressing cells. This readout would potentially also be amenable for detecting msRNA in unselected total PBMCs, but requires further investigation.

References

- 1 Procopio, F. A. *et al.* A novel assay to measure the magnitude of the inducible viral reservoir in HIV-infected individuals. *EBioMedicine* **2**, 874-883 (2015).
- 2 Lungu, C. *et al.* Inter-Laboratory Reproducibility of Inducible HIV-1 Reservoir Quantification by TILDA. *Viruses* **12** (2020).
- 3 Gadkar, V. J., Goldfarb, D. M., Gantt, S. & Tilley, P. A. G. Real-time Detection and Monitoring of Loop Mediated Amplification (LAMP) Reaction Using Self-quenching and De-quenching Fluorogenic Probes. *Scientific Reports* **8**, 5548, doi:10.1038/s41598-018-23930-1 (2018).
- 4 Zerbato, J. M. *et al.* Multiply spliced HIV RNA is a predictive measure of virus production ex vivo and in vivo following reversal of HIV latency. *EBioMedicine* **65**, 103241 (2021).
- 5 Lewin, S. R. *et al.* Use of real-time PCR and molecular beacons to detect virus replication in human immunodeficiency virus type 1-infected individuals on prolonged effective antiretroviral therapy. *Journal of virology* **73**, 6099-6103 (1999).
- 6 Fischer, M. *et al.* Cellular viral rebound after cessation of potent antiretroviral therapy predicted by levels of multiply spliced HIV-1 RNA encoding nef. *J Infect Dis* **190**, 1979-1988 (2004).
- 7 De Scheerder, M. A. *et al.* Evaluating predictive markers for viral rebound and safety assessment in blood and lumbar fluid during HIV-1 treatment interruption. *J Antimicrob Chemother* **75**, 1311-1320 (2020).
- 8 Cynthia, L. *et al.* Analytical treatment interruption: detection of an increase in the latent, inducible HIV-1 reservoir more than a decade after viral resuppression. *medRxiv*, 2023.2011.2014.23298452, doi:10.1101/2023.11.14.23298452 (2023).

REVIEWERS' COMMENTS:

Reviewer #1 (Remarks to the Author):

Thanks to the authors for thoroughly addressing the points raised in the first review, the quality and comprehensibility of the manuscript has significantly increased. Also, I agree with the authors regarding the exon-spanning nature of the LAMP primers. Thanks for detailing this, due to their specific design it was slightly confusing.

However, there are a few minor points that should be addressed.

Although the authors claim at several occasions a correlation between data, these are mostly not statistically significant, which should be stated in the text, e.g.

- line 417-419: the p-value of both, Lungu and Procopio-TILDA, is >0.05

- line 430-432: Please re-phrase the sentence. The correlation of intact DNA with LAMP is not significantly stronger than with total DNA. Instead, the correlation of intact DNA with LAMP is statistically significant, whereas total DNA with LAMP is not.

- line 464: $p=0.516$

- line 533: presumably this refers to Figure 5G, therefore only intact DNA shows statistically significant correlation with LAMP

- line 536: presumably this refers to Figure 5C: This is not statistically significant

line 423: "TILDA_CL" presumably means "TILDA_Lundi"?

Table 1: Please add units in columns for SQuHIVLa and ddPCR DNA (copies per 10^6 PBMCs or CD4+?)

Reviewer #2 (Remarks to the Author):

The authors have done a careful job of revising the manuscript in response to the extensive reviewer comments.

Reviewer #3 (Remarks to the Author):

I appreciate the authors responses to suggestions and manuscript revisions. I have no further substantial comments or concerns.

REVIEWERS' COMMENTS:

Reviewer #1 (Remarks to the Author):

Thanks to the authors for thoroughly addressing the points raised in the first review, the quality and comprehensibility of the manuscript has significantly increased. Also, I agree with the authors regarding the exon-spanning nature of the LAMP primers. Thanks for detailing this, due to their specific design it was slightly confusing.

However, there are a few minor points that should be addressed.

Although the authors claim at several occasions a correlation between data, these are mostly not statistically significant, which should be stated in the text, e.g.

- line 417-419: the p-value of both, Lungu and Procopio-TILDA, is >0.05

As suggested by the reviewer, we rephrased the sentence in the final revised version of the manuscript (lines 432-436) to “The correlation between SQuHIVLa and TILDA_Lungu ($r = 0.62$, $P = 0.0857$, Fig. 5C) was stronger than with TILDA_Procopio ($r = 0.2$, $p = 0.9167$, Fig. 5A) as indicated by Spearman's rank correlation coefficient (r), though statistical significance was not achieved for either comparison, possibly due to the limited sample size ($n=4$) in the SQuHIVLa vs TILDA_Procopio and ($n=9$) SQuHIVLa vs TILDA_Lungu analyses.”

- line 430-432: Please re-phrase the sentence. The correlation of intact DNA with LAMP is not significantly stronger than with total DNA. Instead, the correlation of intact DNA with LAMP is statistically significant, whereas total DNA with LAMP is not.

We rephrased the sentence to “The values of SQuHIVLa showed a significant positive correlation with intact HIV-1 DNA copies ($r = 0.58$, $p = 0.049$), but not with total HIV-1 DNA copies ($r = 0.21$, $p = 0.5199$) (Fig. 5F, G).” in lines 446-448 of the final revised manuscript according to reviewer’s suggestion.

- line 464: $p=0.516$

We rephrased the sentence to “ SQuHIVLa values did not show statistically significant positive correlation with total HIV-1 DNA copies (Pearson $r = 0.2083$, P -value = 0.5160).” in the final revised manuscript (lines 479-480).

- line 533: presumably this refers to Figure 5G, therefore only intact DNA shows statistically significant correlation with LAMP

We rephrased the sentence to “SQuHIVLa demonstrates a significant positive correlation with intact HIV-1 DNA copies but not with total HIV-1 DNA copies” in the final revised manuscript (lines 551-552)

- line 536: presumably this refers to Figure 5C: This is not statistically significant

We rephrased the sentence to “SQuHIVLa exhibited a moderate Spearman’s rank correlation coefficient (r) with the reservoir size measured by TILDA, although this correlation was not statistically significant” in the final revised manuscript (lines 554-556).

line 423: “TILDA_CL” presumably means “TILDA_Lundi”?

We thank the reviewer for pointing this out. It is now corrected to “TILDA_Lungu” in the final revised manuscript (line 440).

Table 1: Please add units in columns for SQuHIVLa and ddPCR DNA (copies per 10⁶ PBMCs or CD4+?)

As per reviewer’s suggestion, we included the units for SQuHIVLa (cells/millions of CD4+ T cells) and ddPCR DNA (copies/millions of CD4+ T cells) in both Table 1 and supplementary table 5.

Reviewer #2 (Remarks to the Author):

The authors have done a careful job of revising the manuscript in response to the extensive reviewer comments.

Reviewer #3 (Remarks to the Author):

I appreciate the authors responses to suggestions and manuscript revisions. I have no further substantial comments or concerns.

We thank all the reviewers for their valuable feedback and constructive comments, which have enhanced the quality and clarity of the manuscript.